# Revealing schoolchildren's key situations in the use of digital media inside and outside school: A media diary study

**Jennifer Virginie Meier** *, **Kai Kaspar**

Department of Psychology, University of Cologne, Cologne, Germany

* j.meier@uni-koeln.de

## Abstract

### Introduction

Digital media have become integral to schoolchildren's lives, both within educational and non-educational settings. Educators emphasize the importance of bridging the gap between school learning and children's out-of-school activities. To identify potential variations and commonalities, we investigated key situations with digital media among lower secondary schoolchildren in Germany, aiming to determine which themes are especially relevant in different settings.

### Methods

We analyzed the media usage of German schoolchildren (ages 10 to 17) in class and outside of school using a mixed-method approach with a focus on the qualitative facets of key situations. For this purpose, 49 schoolchildren from seven schools were asked to complete media diaries. Over a six-week period, they documented key situations with details on setting, emotional experiences, post-communication, social support, and self-reflection. We analyzed the key situations using content and frequency analyses.

### Results

The schoolchildren reported 145 key situations, from which we developed a categorization system comprising 15 distinct categories. The most reported key situations involved "playing video games", "digital learning or homework", and "online communication or content sharing". Most key situations (115) took place outside of school, while 30 occurring in class. Schoolchildren mostly reported key situations that involved feelings of happiness and low levels of arousal. They discussed key situations more often and in greater depth with family members and school friends, and less often with school staff or non-school friends. Key situations were discussed more often face-to-face than online. Notably, schoolchildren engaged in self-reflection on their key situations, including concerns about excessive media use.

**Data Availability Statement:** All data are available from the Gesis database: http://dx.doi.org/10.4232/1.14225).

**Funding:** This research project was funded by the German Federal Ministry of Education and

Research (BMBF) under the funding code 01JD1829C. The authors are responsible for the content of this publication. The funders had no role in study design, data collection and analysis, decision to publish, or preparation of the manuscript.

**Competing interests:** The authors have declared that no competing interests exist.

## Discussion

Overall, the study provides comprehensive insights into the digital media behaviors of schoolchildren, highlighting individual differences and preferences. The discussion offers valuable implications for both educational practice and future research, particularly regarding the integration of digital media into formal educational settings.

## 1. Introduction

Digital media have significantly increased in importance for schoolchildren in both educational [1, 2] and non-educational [3] settings in recent years. Digital media encompass interactive and customizable content available in digital form, such as eBooks, social media updates, and online videos [4], enabling a level of engagement and accessibility not offered by traditional media. Unlike traditional media, which is more static and limited in interactivity [5], digital media can reach global audiences instantly and allows users to engage in participatory, spreadable, and scalable ways [6]. Digital media include a broad array of electronic gadgets that support the creation, storage, and consumption of digital content, ranging from mobile phones and tablets to laptops, game consoles, and smart devices. In an increasingly digitized world, schoolchildren are surrounded by these digital media. This has implications for various aspects of their daily lives, including their social interactions, education, and leisure activities. Within educational environments, digital media are being utilized as interactive and engaging tools to enhance teaching methods, surpassing the limitations of traditional approaches such as reading books, writing on the blackboard, and taking notes in textbooks [2]. Teachers utilize digital technologies to support and enhance learning processes [7, 8], and to plan and implement schoolchildren-centered learning activities [9]. Beyond the classroom, digital media have seamlessly integrated into the everyday routines of schoolchildren [10]. Schoolchildren use digital media for various purposes, including communication (e.g., social media [11–13], communication apps [14, 15]), entertainment (such as video gaming [13, 16], streaming music [17] or watching videos [13]), information retrieval (e.g., search engines [18], read news online [19]), and creative expression (like creating photos or films [20] or drawing pictures [21]). Digital media provide them with access to diverse resources, facilitate connections with peers, and open new opportunities for creative expression.

Given the increasing presence of digital media in both educational and non-educational settings, it becomes imperative to examine how schoolchildren engage with these technologies. Current international research on the implementation of digital media and media education often focuses on individual educational settings, examining them from the perspective of the responsible educational actors. National studies have consistently found significant deficits in both the technical-organizational aspects of digitalization in formal educational settings and in the application of media pedagogical and media-didactic concepts [22, 23]. Considering these challenges, educators have stressed the importance of bridging the gap between school learning and children's out-of-school activities to support their cognitive and personal development [24, 25]. Comparative studies offer valuable insights that can inform teacher education, fostering the integration of classroom learning with lifelong learning principles [26].

### 1.1 Key situations in media use

One way to investigate the media usage of schoolchildren across different settings is through key situations. Key Situations (or key moments) refer to specific, significant moments or

interactions that hold substantial meaning and impact for individuals involved. These situations often reveal underlying issues, emotions, and behaviors that are critical for understanding psychological dynamics. Research explored these situations in various specific contexts, including weight management [27], relationships between nurses and relatives in intensive care units [28], and in psychotherapy [29]. However, the present study specifically focuses on key situations in which digital media have a substantial impact on the lives and activities of schoolchildren. In the situational approach [30], key situations are identified in which important themes are reflected for schoolchildren [31]. The situational approach to teaching emphasizes creating realistic and meaningful scenarios to enhance learning, while also allowing schoolchildren to introduce their own topics [31]. This encourages active participation and engagement, making the learning process both relevant and dynamic. These key situations must be "emotionally and socially significant to the schoolchildren so that their interest (their being-in-the-world) is engaged and challenged" [32]. Key situations can be important for educational work, as they could be addressed in dialogues with the children [33].

Our investigation followed this situational approach and is theoretically grounded in the Uses and gratifications approach (UGA), which aims to understand the motivations and needs that guide individuals in the selection and use of media services [34]. Models within the framework of the UGA usually begin with the premise that media consumers have specific psychological and societal needs, and associated motivations. These needs generate expectations regarding how media can address those requirements, consequently prompting individuals to adopt specific media usage patterns to attain the desired gratifications [34]. While UGA is not a homogeneous theory [35], it presents a valuable framework for elucidating diverse aspects of digital media usage among schoolchildren, as it focuses on individual needs and the proactive selection of media content by users. To determine which individual needs have been attempted to be satisfied, we formulated the following research question:

**RQ1:** What is the nature of the key situations that schoolchildren experience with digital media?

## 1.2 The setting in which key situations take place

Educators have long recognized the importance of bridging the gap between classroom learning and real-life experiences, to foster children's intellectual growth and overall development [24, 25]. Through the examination of comparative studies, valuable insights can be gained, leading to recommendations for teacher education that aim to facilitate the seamless integration of formal education and lifelong learning [26]. However, limited attention has been given to the convergence of digital media usage in both formal educational settings (i.e., curriculum-based and institutionalized learning opportunities within the school) and informal settings (i.e., outside of school). Previous research has primarily focused on these settings in isolation. First, there are studies that investigated the integration of new technologies to enhance formal learning experiences within school settings [36–38]. Second, there are studies within school settings that concentrated on media education, including the development of digital literacy skills [39–41]. Last, there are studies that examine digital media usage in informal settings, such as surveys on usage duration [42–44], investigations into the (mostly negative) impacts of digital media [45–48], and more recently, numerous studies exploring the correlations between social media and behavior [49–52]. Studies that simultaneously address both formal and informal settings and investigate their differences and similarities are scarce. The aim of our study is to address this research gap and provide novel insights by utilizing a mix-method

approach to explore the media usage of schoolchildren in both educational and non-educational settings. For this reason, we asked the following question:

**RQ2:** In which settings (in class vs. outside of school) did the schoolchildren experience key situations with digital media?

## 1.3 Emotional experiences during key situations

Digital media have the potential to evoke positive emotions like enjoyment and excitement, which can be leveraged for educational purposes. For example, digital games designed to enhance social-emotional skills can motivate children to learn by incorporating humor and creative elements [53]. Furthermore, everyday smartphone use has been linked to positive emotional experiences, suggesting that digital media can play a role in managing and improving emotional well-being [54]. In addition, digital communication can sometimes improve the mood and encourage health-promoting behaviors [55]. However, despite these benefits, the use of digital media can also contribute to increased anxiety and depression in children, particularly because of negative social comparisons, inadequate emotion-regulation skills, and exposure to cyberbullying [55].

According to UGA, individuals are more likely to revisit a particular medium if their expected gratifications are met, indicating that media consumption is driven by the alignment of media use with anticipated needs [56]. Conversely, if a schoolchild chooses a medium that does not meet their current needs, media use may lead to a less satisfying experience. Moreover, the Mood Management Theory [57] delves into the intricate relationship between media consumption, emotions, and valence. This theory explores how individuals actively select media content to manage their emotional states and achieve specific mood alterations, based on the premise that media consumption is intricately connected to emotional experiences [57]. Scott and colleagues [58] demonstrated in this context that adolescents using digital technologies for emotion regulation often experience temporary relief from negative emotions, but this is followed by increased sadness and loneliness the next day.

In the realm of media reception research, the measurement of valence and arousal has become a standard practice for assessing the emotional state following media usage [59, 60]. Emotion is conceptualized in the literature as having multiple dimensions, with arousal and valence being key components [61]. Arousal, described as a continuum ranging from calm or indifferent to excitement, represents a spectrum of intensity, while valence reflects a continuum from positive to negative, with neutrality positioned in the middle [62, 63]. Numerous theoretical models posit the independence of valence and arousal from each other [64–66].

Understanding how schoolchildren perceive their media usage emotionally provides deeper insights into their digital media behavior. Through this exploration, our goal is not only to comprehend their experiences but also to lay a solid foundation for better capturing the interplay between emotional needs, media usage, and individual behavior. To delve into these dynamics, we examined key situations guided by the following research question:

**RQ3:** How did schoolchildren emotionally experience their key situations with digital media in terms of valence and arousal?

## 1.4 Post-communication about key situations and social support

The situational approach posits that communication with educational caregivers should take place after key situations, allowing schoolchildren to (post)process them [33]. As the primary

link between people, communication enables the sharing of information, the motivation of actions, and the expression of feelings. In particular, communication between parents and children is crucial from an early age and influences several psychosocial outcomes at the child level. This influence encompasses academic factors like school readiness and performance [67], socio-cognitive factors such as self-esteem and self-development [68], as well as socio-relational factors like peer competence and conflict management [69, 70]. Interacting with school friends at school is also crucial. Positive peer relationships, especially reciprocal friendships, play a pivotal role in fostering development [71]. In the context of peer play, children engage in active dialogue, sharing thoughts and ideas, which serves as a platform for refining oral language skills and developing perspective-taking abilities [72]. During play, children acquire knowledge from their peers through responses to behavior, direct instruction, or assistance [73]. Therefore, schoolchildren could potentially discuss their key situations with various significant individuals, including parents, family members, guardians, or peers. However, Zimmermann et al. [74] found that children and adolescents rarely sought to discuss the content of political YouTube videos with others. When they did engage in discussions, they preferred communicating with their friends rather than teachers. Younger age groups more often talked with their family members about the YouTube videos compared to those aged 20 and above. Furthermore, across all age groups, face-to-face communication was the most common method of discussing the videos, as opposed to other online channels [74]. Although there is an ongoing trend towards computer-mediated interpersonal communication, even among children [75] and reinforced by the COVID-19 pandemic [76], face-to-face communication with peers and adults remains crucial even after the pandemic [77, 78]. To understand how communication unfolds in key situations and how schoolchildren feel supported, we formulated the following research question:

**RQ4:** With whom, how (online vs. face-to-face), and how intensively discuss schoolchildren their digital key situations and how supported did they feel?

## 1.5 Reflection on key situations

Reflection plays an important role in education, discussed by scholars from different perspectives and for different purposes. Kolb's experiential learning model [79], for instance, emphasizes reflection as the catalyst for learners to convert concrete experiences into abstract concepts. It empowers learners to extrapolate main ideas, principles, and abstract concepts from their experiences [79]. Moreover, within the realm of learning, reflection is crucial for students to revisit and enhance their understanding of what they have learned [80], fostering a heightened awareness of their continuous learning journey and skill-building process [81]. Additionally, as demonstrated by Rüth and Kaspar [82], reflection in the classroom can be specifically facilitated with the aid of digital technologies such as video games.

Reflection also plays a significant role in the UGA, as the selection of media is driven by motives that are sought to be satisfied [34]. These sought gratifications represent the anticipated benefits that individuals hope to derive from their media consumption. However, the actual satisfaction of these sought gratifications, referred to as obtained gratifications, is influenced by the reflective processes that follow media engagement. The significance of reflective processes in evaluating whether the intended benefits were met or not becomes even more apparent. These processes not only promote individual growth, but also serve as invaluable tools for understanding, interpreting, and coping complex situations. We asked the following research questions to investigate the reflection processes of the key situations:

**RQ5:** What did schoolchildren learn from their key situations and what would they do differently in hindsight?

## 2. Material and methods

The study's data is derived from a comprehensive research project conducted in Germany that examined media education processes in lower secondary schools, which typically include grades 5 through 10. As part of this project, we explored the use of digital media by schoolchildren in school and outside of school across eight public schools. This led to the development of two distinct media diary studies, which shared a common initial survey. Apart from the shared initial survey and the same recruitment process, the studies had no further commonalities. In the first study [83], we surveyed children from a single school that was not included in the present study. In this research paper, we present findings from the media diaries with the focus on key situations, including data from schoolchildren from the other seven schools. The data set can be accessed [84]. The study received ethical approval from the ethics committee of the faculty of human sciences at the University of Cologne (KKHF0106).

### 2.1 Sample

We created the survey with the software Unipark (Tivian XI GmbH). Initially, all school committees agreed to allow schoolchildren their participation in the research project. The schools provided information to their schoolchildren about the opportunity to participate in the study. Registration for the study was conducted exclusively by parents or guardians, who provided informed consent in writing as part of the online registration process for their minor schoolchild's participation. They were informed that participation was voluntary and without any incentives. They were assured that their children would not be disadvantaged if they chose not to participate, that they had the right to withdraw at any time, and that strict adherence to data protection guidelines would be ensured. If parents or guardians agreed, they provided either their own or their child's email address for the weekly contact. To obtain as diverse and heterogeneous responses as possible, we assigned schoolchildren from a total of seven different schools in the present study, including five grammar schools, a secondary school, and a comprehensive school in urban and suburban locations. Two of the schools follow a structured full-day program, while the others offer flexible options, with all schools hosting between 600 and 1,100 schoolchildren. Only schoolchildren who participated in at least one measurement point were included in the data analysis, resulting in 23 schoolchildren classified as dropouts. Notably, two of them attended the study but reported they had not experienced any key situations and subsequently dropped out. There were no differences between dropouts and participating schoolchildren in terms of age and the Big Five personality traits (all $ps > 0.287$), nor were there notable differences in gender, school affiliation, or grade level. The final data analysis included a total of $n = 49$ schoolchildren (31 female, 18 male). Out of them, $n = 23$ participated at all four weekly measurement time points, $n = 8$ participated three times, $n = 11$ participated two times and $n = 7$ participated only once. The schoolchildren were between 10 and 17 years old ($M = 11.82$, $SD = 1.59$) and in 5th to 10th grade (5th grade $n = 14$, 6th grade $n = 9$, 7th grade $n = 9$, 8th grade $n = 13$, 9th grade $n = 3$, higher than 9th grade $n = 1$). The Big Five personality traits observed in the schoolchildren show expected individual differences. However, the means of these traits were similar to those found in previous studies on children [85] and young adults [86], indicating that the sample reflects relatively typical characteristics of the population.

## 2.2 Procedure

On October 15, 2021, we sent the link to the initial survey along with a personal code to the contact address, and the study ran for 27 days. Upon completion of the initial survey, we began disseminating the media diary. Over a six-week period, we sent the access link to the weekly media diary every Friday at noon, and schoolchildren could complete the study until Monday morning to participate. This procedure aimed to prevent retrospective completion of the media diary and minimize bias due to false memory [87]. The schoolchildren were asked to complete the media diary four times, after which they were not invited to participate again. However, they were allowed to miss measurement points during the six-week period without dropping out of the study. Therefore, participation in the study was measured at different times over the six weeks. The first measurement began on November 5, 2021, and the final weekend was on December 19, 2021. We intentionally chose this time frame to ensure that no major events such as school holidays, exam periods, or other significant disruptions occurred. Each of the four media diary surveys was structured identically and referred to the previous school week. Prior to each measurement time, the schoolchildren were required to provide their unique four-digit code, which allowed for data matching. Afterwards, they were informed about their rights and asked to provide explicit consent to participate.

## 2.3 Questionnaire

In the initial survey, we asked for relevant demographic information, such as age, gender, school affiliation, grade, and whether they participated in any specific non-formal activities at school. Notably, other variables, such as self-regulation, Big 5 personality traits, intrinsic and extrinsic motivation, and self-assessed digital literacy, which were collected as part of the larger research project, were not examined in detail in this study. The weekly diary focused on key situations of media use:

First (RQ1), we asked the schoolchildren about the nature of their key situations with digital media this week. We asked, "Did you have a situation with digital media this week that stuck in your mind?" and elaborated that this could include anything they did particularly well, something they were especially proud of, or even something that upset them or did not work out as planned. The schoolchildren were then asked to formulate this moment in a few sentences, with the constraint that their response should not exceed a maximum of 500 characters. After this, we indicated that all further questions will be about the described situation.

Second (RQ2), we asked where the schoolchildren were in this situation with the options "in class" and "outside of school". Additionally, we inquired about which digital media they used in their key situations, with options including smartphone, tablet, computer, laptop, gaming console, television, and others (open-ended). Regarding the software used, the schoolchildren could openly name up to five programs, apps, or websites.

Third (RQ3), schoolchildren rated their emotional reaction to the key situation in terms of valence and arousal. This was measured with the Self-Assessment Manikin scale (SAM [88]). To assess the intensity of experienced valence, the presented 5-level picture motifs should be assigned to the emotions "happy" (= 5) to "sad" (= 1). Regarding the intensity of emotional arousal, the pictures were assigned to "excited" (= 5) to "indifferent" (= 1) based on the 5-level motif of an exploding manikin.

Fourth (RQ4), we explored post communication about the key situation by asking the schoolchildren if they talked to anyone about their individual key situation. If so, additional inquiries regarding communication were posed; if not, these questions were automatically bypassed. We asked how intensively (0 = "not at all" to 4 = "very intensively") they talked about the situation with different groups of people (school friends, non-school friends, family

members, school staff), whether the communication was online or offline. To measure how social support was perceived during the key situation, we asked the schoolchildren: "How much did you feel supported by other people during the situation?" on a scale from 0 (= not at all) to 4 (= very much). Additionally, there was an option to select "not applicable to the situation".

Last (RQ5), we inquired the schoolchildren about reflection processes. We specifically asked whether they would make any changes in retrospect and what they learned from the situation. Both questions could be answered openly with a word maximum of 250 characters. In the coding process, responses related to changes and learning were categorized into two categories (would change something vs. would not change something; learned something vs. learned nothing).

## 2.4 Data analysis

A mixed-methods approach was used to investigate schoolchildren's key situations with digital media. We analyzed the qualitative data using Mayring's standard content analysis approach [89] and following our previous work [90]: We read all qualitative statements and condensed them to capture each statement's core message. To develop a robust category system, we initially worked inductively, identifying patterns and themes through an in-depth review of approximately 10% of the data. This preliminary coding phase allowed us to construct initial categories broad enough to encompass diverse experiences but specific enough to differentiate among types of media usage and key situations. Next, two independent raters coded a subset of the material, calculating inter-coder reliability to identify potential inconsistencies and refine category definitions. Through this iterative adjustment process, we optimized the category system to ensure common quality criteria, namely accuracy, exclusiveness, and exhaustion [91]. The final category system was then applied to code all data, providing a structured, flexible framework for capturing the full range of schoolchildren's digital media experiences. Using the final category system, all data were coded by the raters, and inter-coder reliability was computed by using Kappa [92], showing an excellent agreement (κ = .96). In rare cases of disagreement between raters, a mutually agreed solution was subsequently found via discussion between these two raters. There was an additional residual category for key situations that did not fit into any of the other categories. To analyze differences between in class and outside of school, we calculated $t$-tests for valence and arousal. All other calculations and comparisons were evaluated descriptively due to the nature of the data and their inappropriateness for inferential testing.

# 3. Results

## 3.1 Key situations in media use (RQ1)

A total of 31 girls and 18 boys reported their key situations. Over the six weeks, they reported 145 key situations, with girls contributing 98 of these instances. We coded these key situations into 14 categories, along with one additional residual category, see Table 1. The category with the highest number of mentions was "Playing video games" (21 key situations). This category encompasses all key situations related to playing with digital media or video games. Twelve of these key situations related to video gaming were reported by girls, and nine by boys. Concerning the grade level, it is evident that particularly the schoolchildren from younger grade levels had key situations on this topic (Grade 5 = 10 key situations, Grade 6 = 6 key situations). Four examples follow, with additional statements presented in Table 1:

1. *"Yes, on the PS4, I have to fulfill a few requirements in Fortnite."*, boy, 10 years old, 5th grade.

**Table 1. Overview of key situations: Categories, sample statements, number, adjusted number, and setting.**

| | Category | Sample statements | Total | | Setting | |
|---|---|---|---|---|---|---|
| | | | $n$ | $n_{corr}$ | In class | Outside of school |
| | | | | | $n$ | $n$ |
| 1. | Playing video games | "I played Fortnite for the first time at a friend's place. It was actually fun. I had watched Fortnite before, but playing it myself was more enjoyable.", "I played on the tablet as usual." | 21 | 13 | - | 21 |
| 2. | Digital learning or doing homework | "I created study sheets for my exams on my tablet. I saved them and now I am using them to prepare for my upcoming exams in the coming weeks.", "I did homework on my tablet." | 18 | 12 | 1 | 17 |
| 3. | Online communication or content sharing | "I am sick and use my laptop every day to write with my teachers.", "I've been writing with my friends and sharing about everyday life." | 17 | 16 | - | 17 |
| 4. | Creating, editing, or presenting Office documents | "I made a PowerPoint presentation with a friend in German class.", "I made a spreadsheet about a program in a subject at school." | 14 | 12 | 9 | 5 |
| 5. | Searching for information on the internet | "In politics class, I used my computer to do research.", "I looked up information about elephants on my phone." | 12 | 10 | 6 | 6 |
| 6. | Experiencing and addressing challenges in digital media use | "I was annoyed that my grandma changed the password and that is why I could not restore my game.", "Discussion about WhatsApp security." | 10 | 9 | - | 10 |
| 7. | Watching videos online | "I watched a 'Grip' episode that was exciting", "I watched a bit of YouTube on my phone." | 9 | 8 | - | 9 |
| 8. | Creating digital media products | "I am supposed to film myself making a hot chocolate from chocolate for art class.", "I drew a beautiful picture on my iPad with the Apple pencil." | 8 | 5 | - | 8 |
| 9. | Passing time with digital media | "I was passing the time with all kinds of gadgets", "My parents were away from Saturday to Sunday, and I could do whatever I wanted." | 7 | 5 | - | 7 |
| 10. | Using school servers | "This week I used my school manager more.", "IServ, that's how I found out we were writing an English test." | 6 | 6 | 1 | 5 |
| 11. | Digital documentation and note-taking | "This week, I used my iPad the most at school to take notes (in most subjects).", "I have worked with my iPad in different subjects, as a substitute for a notebook." | 6 | 4 | 5 | 1 |
| 12. | Acquiring digital media competencies | "I transferred photos from a phone to the computer and learned a few things along the way.", "I figured out how to select people on WhatsApp in the status that I do not want to see my status." | 5 | 5 | 2 | 3 |
| 13. | Participating in video conferences or digital workshops | "I had a digital workshop all day Friday. It took place via a laptop. The workshop was also about media and the Internet.", "We had a video conference." | 5 | 4 | 3 | 2 |
| 14. | Using digital media for unspecific educational purposes | "It was fun, and we used digital stuff in school.", "I started doing more for school on the PC. I feel that this way, I learn much more, and I can organize myself better than just holding a book in my hand and reading through the material." | 4 | 4 | 2 | 2 |
| 15. | Residual category | "I ordered certain things this week for my dog he needed new play stuff, food and so on.", "My entry in the youth research competition." | 3 | 3 | 1 | 2 |
| | **In Total** | | **145** | **116** | **30** | **115** |

**Note.** Column $n_{corr}$ shows the number of schoolchildren who provided at least one statement of the respective category (i. e., number corrected for multiple key situations of individual participants falling into the same category)

2. *"At the moment, everything is the same as before regarding the media. However, I discovered a new game on the PC that I really like and that distracts me from some difficult situations."*, girl, 13 years old, 8th grade.

3. *"My friend and I played video games. This is special for me because I don't have any."*, girl, 10 years old, 5th grade.

4. *"I managed to reach FIFA Weekend League with my team, and from the rewards, I pulled the top player 'Salah'. That was truly the best moment of this week."*, boy, 12 years old, 7th grade.

We could assign a slightly smaller number of key situations to the category "Digital learning or doing homework" (18 key situations). This category encompasses all key situations in which schoolchildren reported learning activities. Since it was sometimes not possible to differentiate whether they were studying for homework or something else, the category was expanded to include mentions related to homework. Half of these key situations were reported by boys and the other half by girls. Also, an even distribution was observed across grade levels, with most key situations reported in the 8th grade (7 key situations). Four examples from this category were:

1. *"I have several exams in a few days and am preparing for them with my iPad."*, girl, 11 years old, 6th grade.

2. *"I have a new learning app on my phone that I use regularly now."*, boy, 13 years old, 8th grade.

3. *"I had to study for my math test, and for that, I had to use YouTube multiple times as well as other websites on the internet."*, female, 17 years old.

4. *"Studied English using [the learning app] 'Anton' in preparation for the exam."*, female, 10 years old, 5th grade.

The third most common category of key situations was „Online communication or content sharing" (17 key situations). This category includes all key situations related to communicating with others via digital media and the digital exchange of information or files. This also encompasses situations involving problematic communication, such as conflicts or bullying. Only three key situations in this category were reported by boys; all others were reported by girls. The themes of the key situations varied widely, here are four examples:

1. *"I am sick at home and use my laptop every day to communicate with my teachers and study."*, boy, 10 years old, 5th grade.

2. *"In the class group on WhatsApp, the children in my class insulted each other."*, girl, 12 years old, 7th grade.

3. *"I received funny pictures from my friends."*, girl, 10 years old, 5th grade.

4. *"I really wanted to know the name of the song for which my dance group was rehearsing a choreography. The dance teacher sent me a link to the song via WhatsApp."*, girl, 10 years old, 5th grade.

With 14 key situations, "Creating, editing, or presenting Office documents" is the fourth most frequent category, emphasizing the varied use of office programs from different software providers (such as Microsoft, OpenOffice). Example statements for this and the following categories can be found in Table 1. This is followed by the category "Searching for information on the internet" with twelve key situations. In the category "Experiencing and addressing challenges in digital media use" (10 key situations), various challenges or issues encountered by schoolchildren are addressed. They were able to solve some of these challenges, as in the statement, "*I had to log in, and it did not work at home. When I did it with another person, it worked.*". Others, however, remained unresolved, as in the statement, "*I was annoyed that my grandma changed the password, and that is why I could not restore my game.*". Subsequently, there are key situations related to the categories "Watching videos online" (9 key situations) and "Creating digital media products" (8 key situations). The latter category includes all creative key situations where schoolchildren digitally produced something unrelated to Office programs. Only a few key situations were mentioned in the categories "Passing time with digital

media" (7 key situations), "Using school servers" (6 key situations), and "Digital documentation and note-taking" (6 key situations). The category of key situations "Acquiring digital media competencies" includes five statements in which schoolchildren explicitly wrote about learning something about digital media. The last two categories encompass key situations related to "Participating in video conferences or digital workshops" (5 key situations) and "Using digital media for unspecific educational purposes" (4 key situations). As the schoolchildren reported multiple key situations over the weeks, we also provided the adjusted values of the key situations in Table 1. In some instances, there were notable differences, as in the most frequently reported category "Playing video games", where the number of de-duplicated key situations was reduced to 13. A similar outcome was observed in the "Digital learning or doing homework" category ($n$ = 12). Key situations of "Online communication or content sharing" exhibited a more diverse distribution among the schoolchildren ($n$ = 16). In the remaining categories, almost no differences were found after removing duplicate entries.

To facilitate the interpretation of inter-personal differences in key situations and the temporal fluctuation at an individual level, we created Fig 1. This figure illustrates and summarizes all key situations of the 49 schoolchildren, including information on participants' gender and age, and circumstances of the key situations (category, context, experienced valence and arousal, type of digital media used, communication with others, support received from others). This kind of visualization allows a more nuanced interpretation beyond averages, enabling individual examination of each schoolchild and key situation. We observed that certain schoolchildren encountered consistent key situations throughout multiple weeks. Conversely, some other schoolchildren described diverse key situations from week to week, exhibiting varying themes each time. In Fig 1 it can be seen, for example, that child 1 (female, 10 years old) experienced the first key situation related to "Using school servers" and the three subsequent key situations related to "Playing video games" (column K). A similar pattern is evident for child 10 (female, 11), who also experienced the last three key situations related to "Playing video games". In contrast, child 3 (female, 10) reported all four key situations on different topics, as did child 7 (male, 10).

In summary, the data reveal a high prevalence of digital gaming, learning and communication among schoolchildren when considering the overall count of key situations. However, on an individual level, there is substantial variability in experiences, with certain schoolchildren reporting a dominant theme over several weeks, while others report qualitatively varying key moments over time.

## 3.2 The setting in which key situations take place (RQ2)

To uncover differences between formal and informal settings, the schoolchildren were asked where they experienced their key situations. The results show that in total, $n$ = 115 key situations were experienced outside of school, while only $n$ = 30 key situations took place in class, see Table 1. Key situations from six categories occurred exclusively outside of school, namely "Playing video games", "Online communication or content sharing", "Experiencing and addressing challenges in digital media use", "Watching videos online", "Creating digital media products", and "Passing time with digital media". The only two categories where schoolchildren actively engaged in at least 50% of the experienced situations during class were "Creating, editing, or presenting Office documents" (9 out of 14 key situations) and "Searching for information on the internet" (6 out of 12 key situations). In the least frequent five categories ("Using school servers", "Digital documentation and note-taking", "Acquiring digital media competencies", and "Participating in video conferences or digital workshops") a mixed pattern emerged between experiences in class and outside of school, see Table 1.

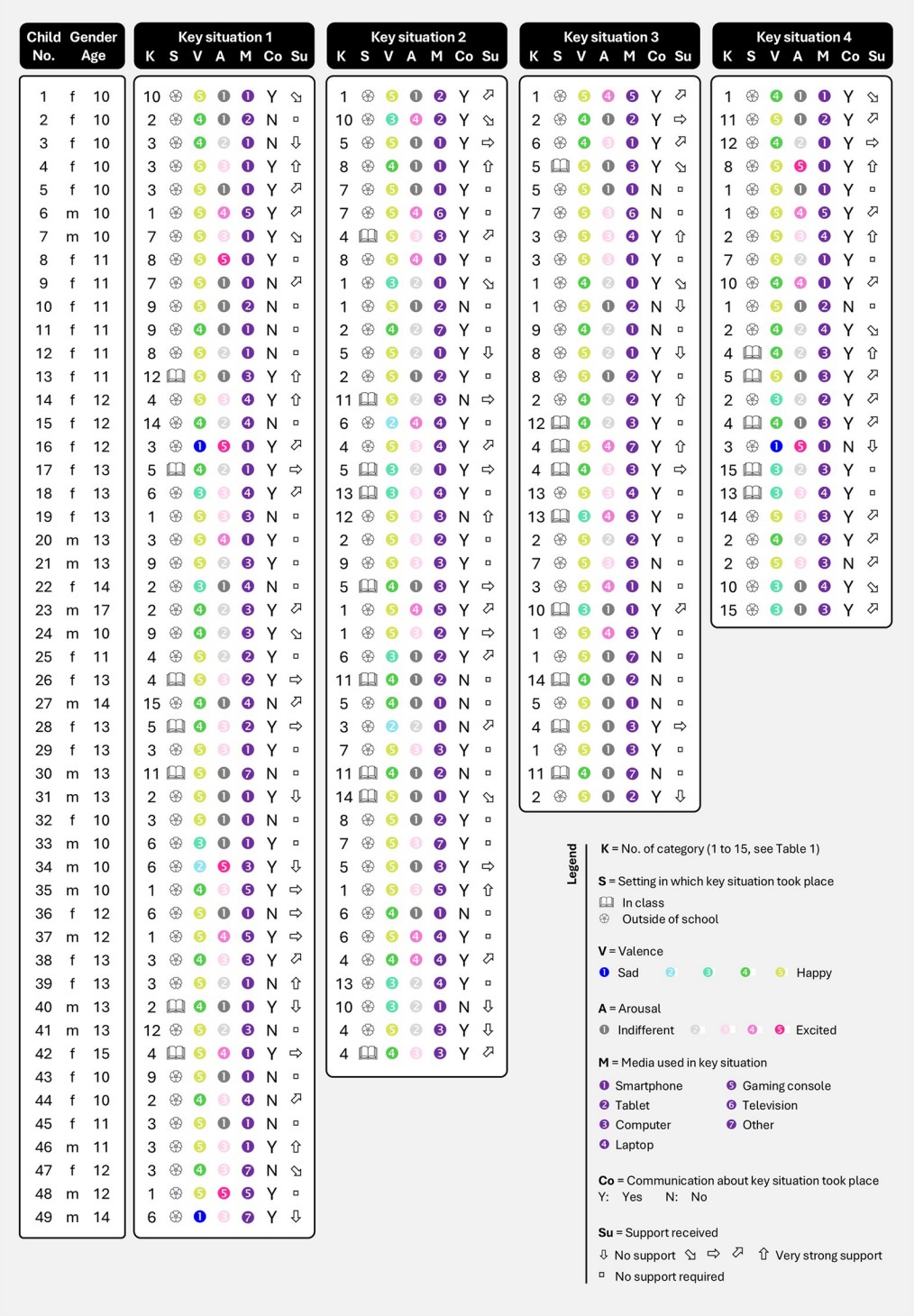

**Fig 1. Overview of all perceived key situations of 49 schoolchildren, including gender, age, and circumstances.**

Analyzing the data on an individual level provides additional insights, as shown by Fig 1. We can identify three types of schoolchildren: The majority of schoolchildren ($n = 31$, 63%) exclusively reported key situations outside of school (e.g., Child 1, 2, 3, 5). Some schoolchildren ($n = 14$, 29%) reported key situations both outside of school and inside the classroom (e.g., Child 4, 7, 12, 13). And a small group of $n = 4$ (8%) schoolchildren (Child 17, 26, 30, and 42) experienced key situations exclusively in class. Examining these four children individually, it becomes evident that Child 30 (male, 13) reported key situations related to "Digital documentation and note-taking" three times consecutively in class, while Child 42 (female, 15) exclusively experienced key situations pertaining to "Creating, editing, or presenting Office documents" in class. Child 17 (female, 13) also encountered only key situations in class, albeit with variations, similar to Child 26 (female, 13), see Fig 1.

Overall, from the perspective of schoolchildren, they encountered a greater variety of key situations outside of school compared to within the classroom. Traditional experiences related to Office documents and information searches were increasingly reported in class as well. On an individual level, the analysis revealed that schoolchildren had very different patterns of experiences with key situations. Importantly, the significance of school for the experience of key moments appears to vary greatly from person to person.

### 3.3 Emotional experiences during key situations (RQ3)

We analyzed the intensity of the schoolchildren's emotional experiences in key situations by examining both valence and arousal. Please note, that lower values indicate lower happiness and lower arousal, respectively. First, we report the results regarding valence. Overall, it appears that the schoolchildren primarily reported key situations in which they were happy ($M = 4.35$, $SD = 0.92$), as the mean valence rating was statistically significant above the midpoint of the scale, $t(144) = 17.75$, $p < 0.001$, $d = 1.47$. The direct comparison between key situations in class ($M = 4.17$, $SD = 0.75$) versus outside of school ($M = 4.40$, $SD = 0.95$) was not statistically significant, $t(143) = -1.24$, $p = 0.216$, $d = -0.26$. In Fig 1, all key situations are individually listed with their corresponding valence indicated by color (blue to green). Here, it is detailed that the whole sample of schoolchildren reported the maximum negative valence (= 1 / sad) for only three key situations, with two of them coming from the same child (Child 16; female, 12) and revolving around the theme of "Online communication or content sharing": *"I was annoyed that so much nonsense was written again in the WhatsApp class group."* The third maximally sad key situation (Child 49; male, 14) was assigned to the category "Experiencing and addressing challenges in digital media use": *"I had a discussion about the safety of WhatsApp".* At the individual level, there were striking differences in the level of fluctuation regarding valence experienced across key situations, with some schoolchildren experiencing only positive valence throughout all four weeks.

In terms of the arousal during key situations, schoolchildren primarily described experiences with relatively low arousal levels ($M = 2.26$, $SD = 1.19$), as the mean valence rating was statistically significant below the midpoint of the scale, $t(144) = -7.51$, $p < 0.001$, $d = -0.62$. The arousal levels of key situations did not differ between outside of school ($M = 2.33$, $SD = 1.22$) and in class ($M = 1.97$, $SD = 1.07$), $t(143) = -1.49$, $p = 0.138$, $d = -0.31$. There was no significant correlation between valence and arousal ($r = -.114$, $p = 0.171$), entirely consistent with the theoretical models that propose the independence of both dimensions. In Fig 1, all key situations are individually listed with their corresponding arousal (gray to pink) indicated. Here it becomes apparent that arousal exhibited a slightly stronger variation among the key situations. Some schoolchildren reported experiencing all key situations as relatively calm to very calm (e.g., Child 5 or Child 30), whereas others expressed heightened excitement across

all of their key situations (e.g., Child 6 or Child 37). Other schoolchildren reported varying arousal levels over the weeks (e.g., Child 4 or Child 9). An example of a very happy and exciting key situation is from Child 4 (female, 10): "*We should take a photo or video of a cocoa for our school calendar on Instagram. My video was the only one from my class shown online*". Child 3 (female, 10) reported an indifferent experience as her second key situation: "*I searched for information about elephants on my phone. And I succeeded without help*".

To sum up, the schoolchildren reported predominantly positive valence during key situations, both in class and outside of school and the emotional experiences were characterized by low arousal levels. However, there were also instances where key situations deviated from this pattern at the individual level, particularly in connection with communication apps.

### 3.4 Post-communication about key situations and social support (RQ4)

In a total of 104 of 145 key situations (72%), schoolchildren engaged in communication with others about their experiences, whereas in 41 instances (28%) they did not communicate with anyone. Most frequently, schoolchildren discussed key situations with family members (e.g., mother, brother, aunt, grandfather. . .), with 97 instances (67%) in face-to-face conversations and only 4 online (3%), as shown in Fig 2. The second most common group with whom schoolchildren sought to discuss key situations was their school friends, with 72 face-to-face

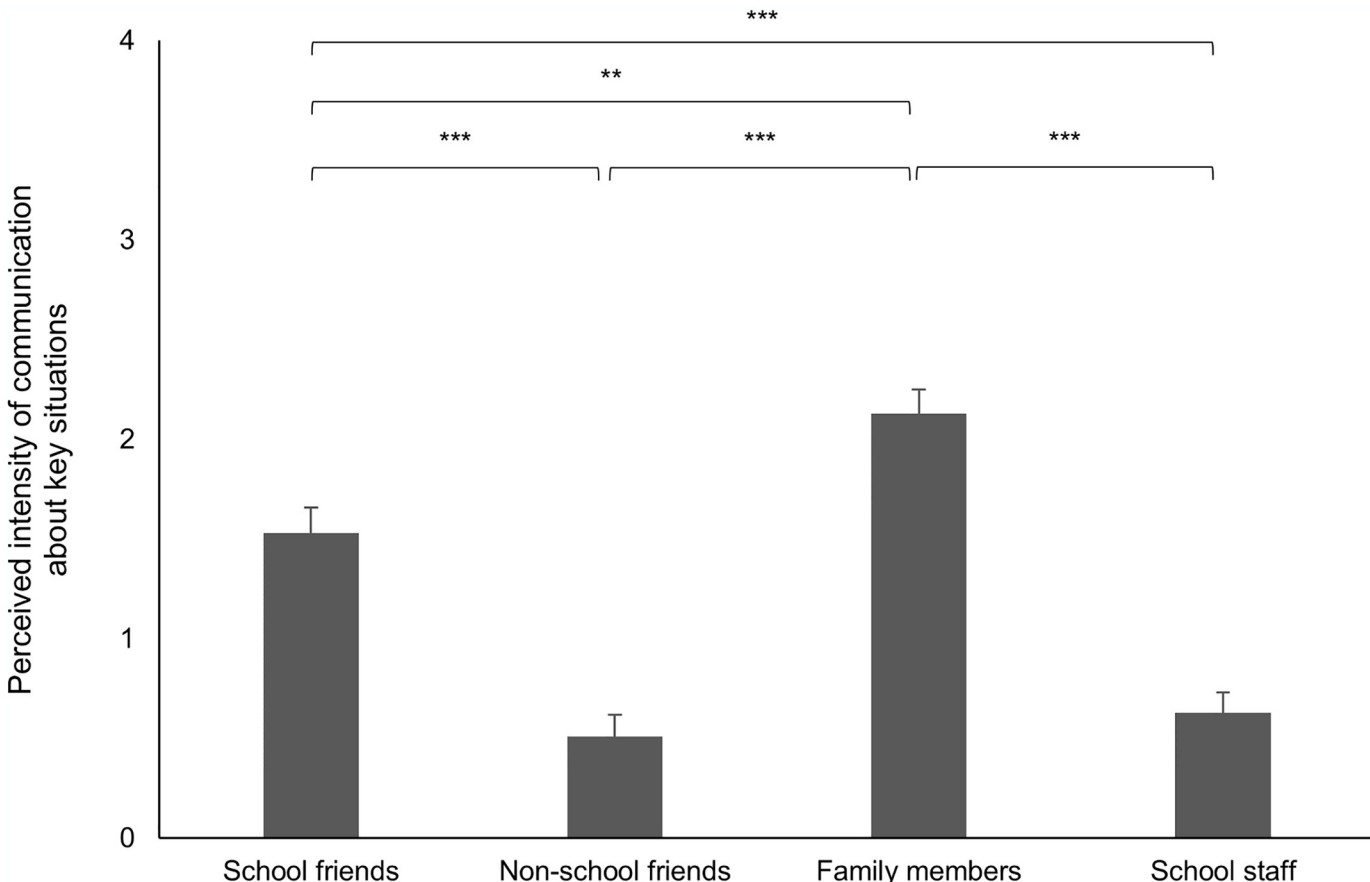

**Fig 2. Number of conversation partners based on *n* = 104 key situations.** Note. Multiple mentions were possible, as conversations about a key situation may have been held with several groups of communication partners through different communication channels.

(50%) and 15 online (10%) discussions. Less frequently, but still in over a third of key situations, they engaged in conversations with school staff (teachers and other school personnel), with 41 face-to-face (28%) and 3 online (2%) discussions. Schoolchildren communicated least frequently with their non-school friends, having face-to-face discussions in 19 key situations (13%) and online discussions in 21 situations (14%) via digital media. Please note that multiple mentions were possible, illustrating that schoolchildren could discuss a key situation with several groups of people. Thus, a discernible preference for direct face-to-face conversations over online communication methods is evident among the schoolchildren, highlighting their inclination toward personal interactions when discussing media-based key situations. In summary, in more than two-thirds of all key situations, schoolchildren sought conversations with others, preferably with their family members and school friends.

Regarding the intensity of conversations, a similar pattern emerges, reflecting the preferences observed in the communication dynamics. As shown by Fig 3, the schoolchildren exhibited varying levels of intensity in their communication with different groups of individuals regarding the key situations. They engaged in the most intense conversations about their key situations with family members compared to all other groups ($M = 2.13$, $SD = 1.23$, all $t$s $\geq 3.18$, all $p$s $\leq 0.002$). Furthermore, they engaged in more intense discussions about their key situations with school friends ($M = 1.53$, $SD = 1.31$) compared to conversations with school staff ($M = 0.63$, $SD = 1.04$), $t(103) = 6.47$, $p < 0.001$, $d = 0.63$, and non-school friends ($M = 0.51$, $SD = 1.11$), $t(103) = 6.17$, $p < 0.001$, $d = 0.61$. Notably, the intensity of conversations

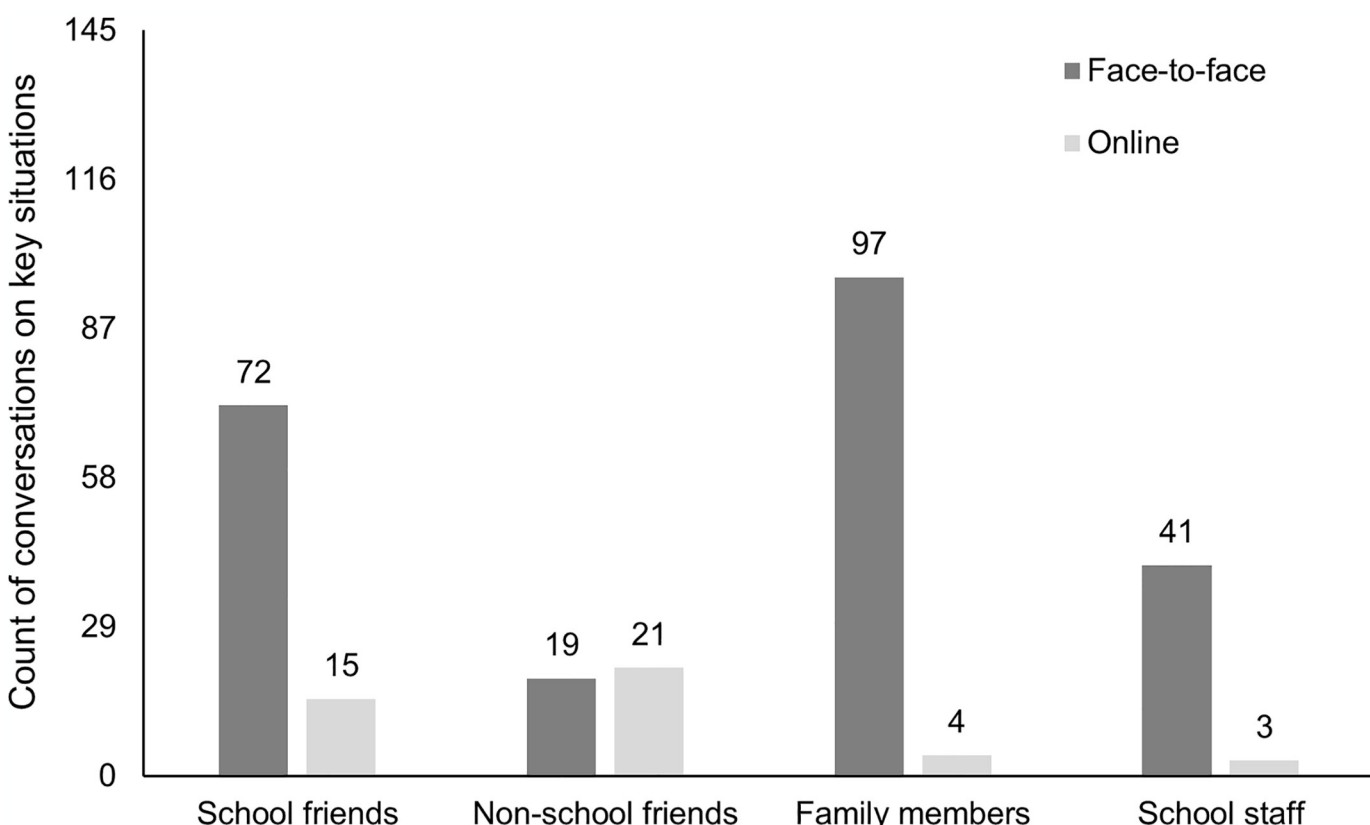

**Fig 3. Intensity of conversations with communication partners ($n = 104$ key situations).** Note. *** $p < .001$, ** $p < .01$. Vertical lines indicate the standard error of the mean.

did not significantly differ between discussions with school staff and non-school friends, $t$ (103) = −0.71, $p$ = 0.479, $d$ = −0.07. Despite the digital nature of these key situations, most schoolchildren preferred discussing them face-to-face with others.

The results regarding perceived social support indicate that in 60 out of 145 key situations (41%), the schoolchildren did not expect or did not need social support during the situation, as they indicated the option "not applicable to the situation". In the other 85 key situations (59%), the schoolchildren received varying levels of support ($M$ = 3.26, $SD$ = 1.29). Fig 1 illustrates how the schoolchildren felt socially supported during their key situations, represented by arrows (column S). For instance, Child 13 (female, 11) strongly felt supported when acquiring digital media competencies in class. Similarly, Child 16 (female, 12) received strong support in the first week despite feeling sad in communication with others. On the other hand, Child 31 (male, 13) felt less supported or not supported at all in his key situations. At the individual level, it is evident that the schoolchildren experienced varying degrees of support in their weekly experiences, with only a few consistently feeling strongly supported (e.g., Child 23, male, 17) or consistently feeling poorly supported (Child 31, male, 13). In summary, our data indicates that in many cases, no social support was expected or requested by the schoolchildren. The quality of social support primarily depended on the individual key situations, with almost all schoolchildren receiving varying degrees of support at different times. Few children experienced consistently strong or weak support.

## 3.5 Reflection on key situations (RQ5)

In terms of the reflection processes on key situations, we considered two perspectives: First, whether the schoolchildren would do something differently next time, and second, what they have learned from the specific key situation.

In 109 out of 145 key situations (75%), schoolchildren would not change anything in hindsight. In 36 key situations (25%), the schoolchildren indicated that they would do something differently, as shown by the complete list of corresponding statements in Table 2. The desired changes varied depending on the key situation and involved different themes. Some conclusions regarding opportunities for improvement revolved around aspects of video games: "*Do not give up immediately if I cannot succeed in something in a video game*" or "*Spend less money on Fortnite*". Others related to aspects of time management, acknowledging excessive media consumption: "*I watched too much TV and did not care about my surroundings. I would do this differently*" or simply "*Play less*". Another set of reflections focused on the individual and how they could behave differently: "*Do not engage with every person*" or "*I would try not to cry in the bullying situation*". Yet others contemplated the program and explored additional possibilities: "*Use a different program*" or "*Save my PowerPoint presentation*".

In 77 key situations (53%), schoolchildren reported having learned something, whereas in slightly less than half of the key situations, they indicated that they had learned nothing. Table 3 shows examples of statements per category. If there are fewer than three statements in Table 3, there were no further statements in this category. The statements can be divided into three contexts. First, the schoolchildren learned things about themselves in various key situations, for example: "*Not to insult children over the internet or at all.*", "*I learned always to be mindful of what I write because it could have consequences.*", "*When you put in a lot of effort, others notice it too*", "*Do not give up*", or "*Playing video games is possible without getting angry*!". Second, they learned something about the digital medium, for example: "*How to edit videos better.*", "*How to make a good presentation.*" or "*There is much more to discover in the digital world than I thought before*". Third, the schoolchildren acquired substantive content or

**Table 2. All 36 statements on what the schoolchildren would do differently in retrospect with regard to their key situations.**

| | Schoolchildren's statements on what they would do differently |
|---|---|
| 1 | Do not give up immediately if I cannot succeed in something in a video game. |
| 2 | Do not get too involved in things. |
| 3 | Learn more. |
| 4 | Do not engage with every person. |
| 5 | Play a bit better, as I only won 2:1. |
| 6 | Buy FIFA 22. |
| 7 | Play less. |
| 8 | Spend less money on Fortnite. |
| 9 | At 'Sofatutor', there is a feature where you can ask questions and get direct answers with explanations, but unfortunately, this feature is only available during the week, and I usually study on weekends. I might have questions on Fridays. |
| 10 | Consider working with an additional app ('Phase6'). |
| 11 | Save my PowerPoint presentation. |
| 12 | I would try not to cry in the bullying situation. |
| 13 | One should not directly engage with it and rather ignore it, but also address it as it is not okay. |
| 14 | I once had a disagreement with a friend via messages. Next time, if possible, I would do it in person because you can see the reactions better. |
| 15 | Insert more videos and sounds. |
| 16 | I watched too much TV and did not care about my surroundings. I would do this differently. |
| 17 | Do not sign up everywhere without thinking. |
| 18 | I would do it directly with another person and read everything more thoroughly. |
| 19 | Press the power button longer earlier. |
| 20 | I would pay more attention and save it in between so that everything is not lost at once. |
| 21 | Wait until December; then I will get a new hotspot. |
| 22 | Eat more popcorn. |
| 23 | I was at home a lot and almost never outside. I would do it differently next time. |
| 24 | Take care of myself more. |
| 25 | Check the school manager every day. |
| 26 | I would copy the entire folder directly instead of scanning each individual image and then copying it. |
| 27 | Sometimes I would prefer to spend more leisure time, but due to school, there is often no time. |
| 28 | It might be better if I engage with it more because I only deal with it once a week. |
| 29 | Include more or different questions. |
| 30 | For questions, directly approach the teacher or work step by step. |
| 31 | I think I need to be more motivated in the situation. |
| 32 | Use a different program. |
| 33 | Engage more with the programs I used. |
| 34 | Do not be too hasty and wait until the teacher explains it to me again or ask questions about the given task. |
| 35 | Enter more specific search terms. |
| 36 | In hindsight, maybe make a PowerPoint presentation. Otherwise, I think I handled it quite well. |

**Note.** The numbers are solely an enumeration of all statements and do not correspond to a category nor to a schoolchild.

procedural knowledge: *"How to indicate the function in the coordinate system. f(Position) is Function value"*, *"More about music"*, or *"I learned a lot about my presentation topic."*.

In summary, the results demonstrated that in most key situations, schoolchildren would not do anything differently in hindsight. However, schoolchildren exhibited critical reflections

**Table 3. A selection of open statements made by the schoolchildren regarding what they have learned from the key situations, sorted by category.**

| | Category | Schoolchildren's statements on what they have learned |
|---|---|---|
| 1. | Playing video games | Stay strong always. |
| | | Playing video games is possible without getting angry! |
| | | Better and more learning, therefore, play less. |
| 2. | Digital learning or doing homework | How to structure my study materials. This way, I can create better study notes over time. |
| | | How to effectively engage in digital learning (productively) without getting distracted. |
| | | What we discussed in class before helped me in the situation. |
| 3. | Online communication or content sharing | That it is not a good idea to handle things over messages, and I should do it in person next time. |
| | | I learned always to be mindful of what I write because it could have consequences. |
| | | Not to insult children over the internet or at all. |
| 4. | Creating, editing, or presenting Office documents | How to create a table. |
| | | How to make a good presentation. |
| | | That I enjoy giving presentations. |
| 5. | Searching for information on the internet | It is good to know and have your login credentials handy. |
| | | I learned that technology evolves incredibly fast. |
| | | I learned a lot about my presentation topic. |
| 6. | Experiencing and addressing challenges in digital media use | I need to look at everything more closely and read through it. |
| | | It is fun to help others. |
| | | That you can learn something new with the help of friends. I now know where the settings are on my phone. |
| 7. | Watching videos online | How to handle the phone. |
| | | I spent more time outside in the next period than before. |
| | | That it is fun to goof around with cars on closed roads. |
| 8. | Creating digital media products | How to edit videos better. |
| | | If you put in a lot of effort, others will notice. |
| | | How difficult it is to film simple things and stage them to look great. |
| 9. | Passing time with digital media | How I can handle it even better. |
| 10. | Using school servers | You cannot catch important things. |
| | | Check IServ more often for information. |
| 11. | Digital documentation and note-taking | Enjoying math in a different situation. |
| | | How to extend a table in Word. |
| 12. | Acquiring digital media competencies | I should copy the entire folder. |
| | | I learned how to make a documentary, how to write it, and how to do good research. |
| | | How PowerPoint works. |
| 13. | Participating in video conferences or digital workshops | - |
| 14. | Using digital media for unspecific educational purposes | You need to have fun sometimes, and you should still learn enough for school for yourself. |
| 15. | Residual category | You can really find everything on the internet. Everything! |

**Note.** If fewer than three statements are listed in the table, this is the complete set of learning-related statements provided in that category.

in one-fourth of all key situations. The depth of reflection varied substantially between critical media consumption and simpler program-related issues such as saving the documents more often or using a different program. In half of all key situations, schoolchildren indicated that they had learned something during the experience, with variations in the depth of their reflections on the topic. These learning experiences ranged from personal growth and self-awareness to digital media skills and academic knowledge. This underscores both the relevance of key situations for learning and the children's awareness of it. However, it is noteworthy that in the remaining half of key situations, either no learning was reported, or it was not perceived as such.

### 3.6 Hardware and software

Finally, for an even more complete impression of the key situations, we analyzed the hardware and software used in these situations. The schoolchildren indicated which digital media they used to experience the key situation and which software (programs, apps, websites) they used. Fig 4 shows the distribution of the 145 key situations across different digital media. The most frequently used device was the smartphone (in 51 key situations), followed by computers (32 key situations), the tablets (28 key situations), laptops (18 key situations), a gaming console

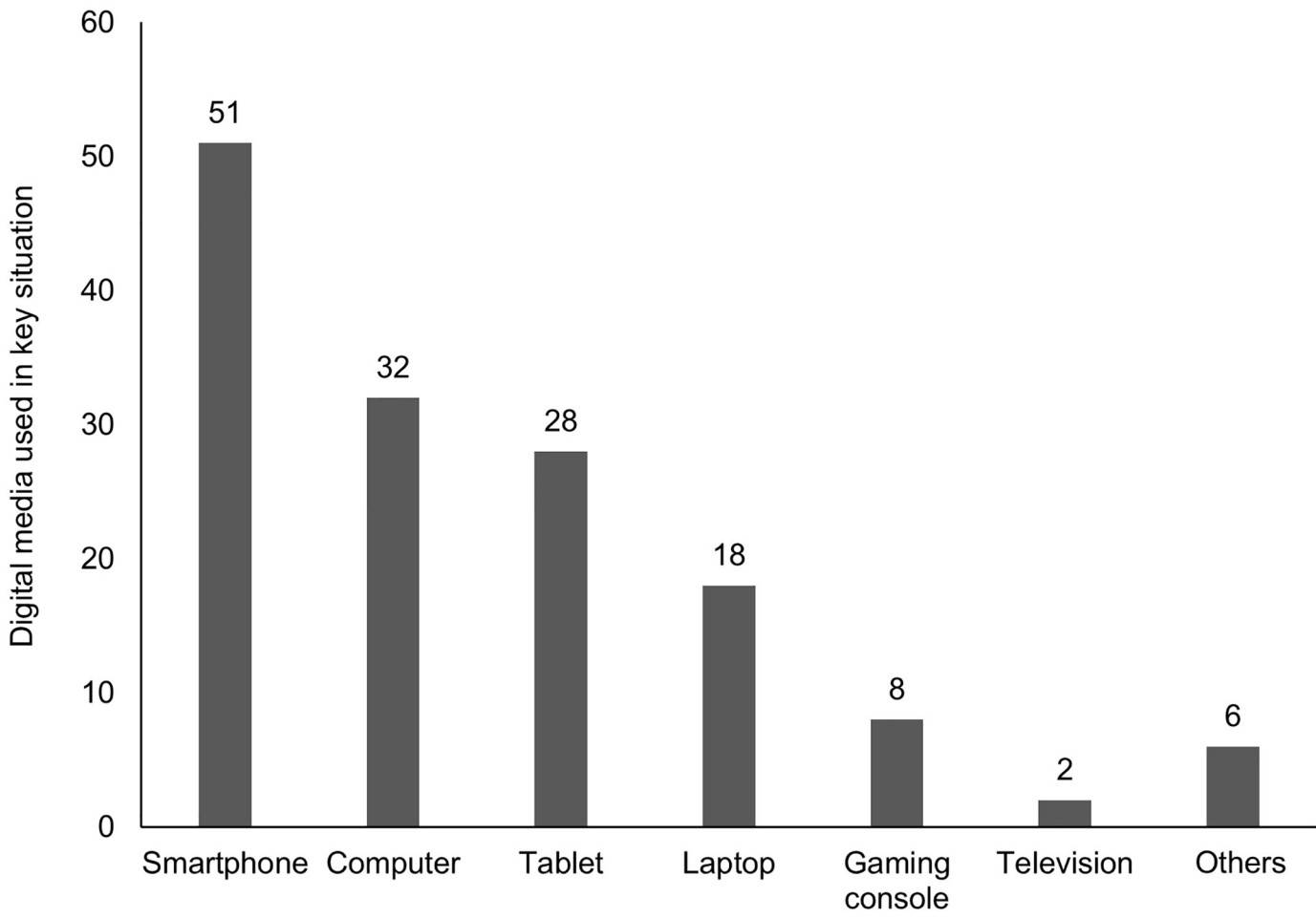

**Fig 4. Digital media used in key situations.**

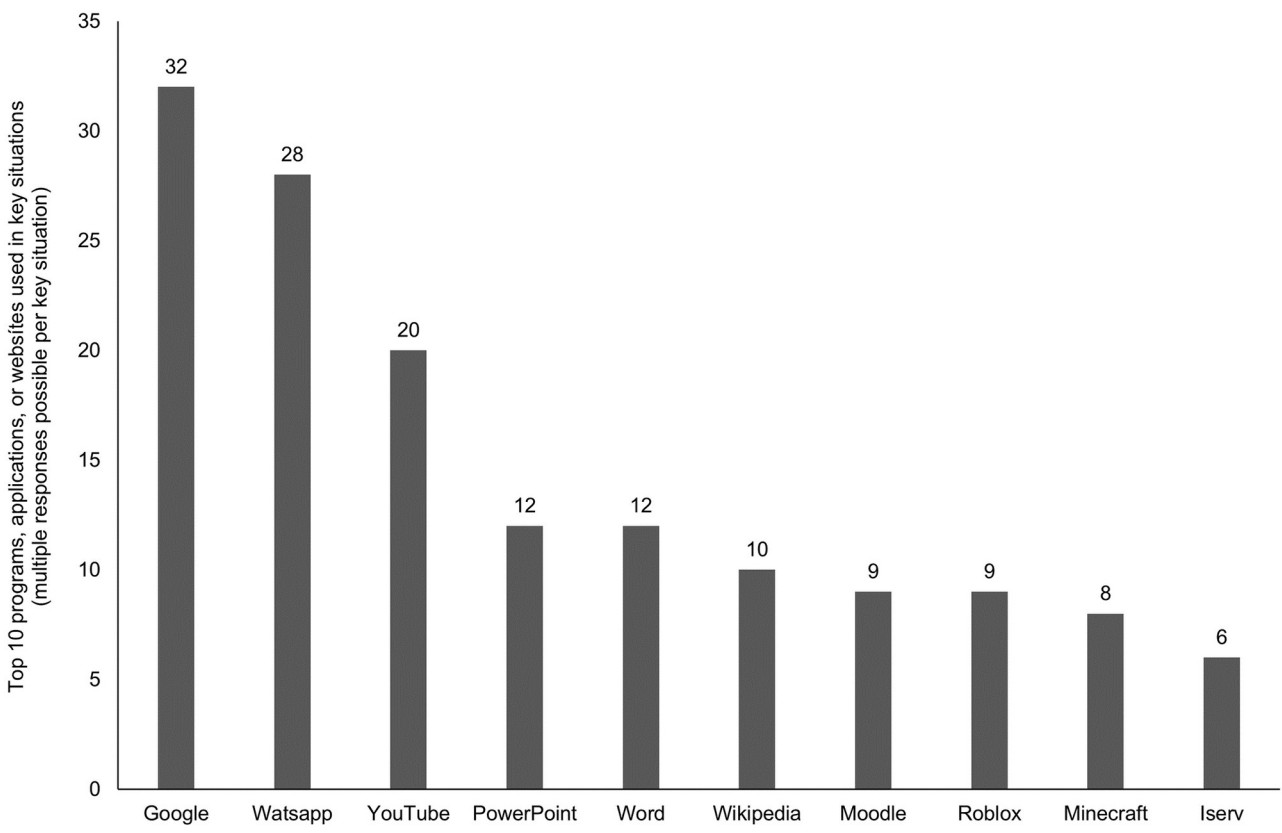

**Fig 5. Top 10 programs, applications, or websites used in key situations (multiple responses possible per key situation).**

(8 key situations), and a television (2 key situations). In six cases, another medium was used (two cases with smartboard and multiple devices and one with television), including one case where the topic was related to digital media but was discussed face-to-face without using any device. Regarding the software used in the key situation, the schoolchildren could mention up to five items per situation. The top 10 of these are presented in Fig 5. The most frequently used software was Google, followed by WhatsApp, YouTube, and two Office programs and Wikipedia.

## 4. Discussion

We conducted a comprehensive examination of the media behavior of schoolchildren, aiming to explore their key situations with digital media in both formal and informal settings. Initially, all results of the research questions are presented consecutively, discussed, and the implications are provided for each domain. Following this, limitations of the study are outlined, followed by the conclusion.

### 4.1 The nature of key situations

In this study, key situations were understood as situations in which digital media have a significant influence on the lives and activities of schoolchildren. To identify the motives and needs those schoolchildren had in their media usage, we coded the qualitative statements of the schoolchildren according to Mayring [89] into 15 categories. Consistent with a representative

survey by MPFS [93] for 12- to 19-year-olds, (video)gaming, entertainment, and communication played a central role in schoolchildren's media usage, which was also evident in the study by Meier and Kaspar [83]. Additionally, many key situations arose during learning or while doing homework. There were two groups of schoolchildren, namely those who experienced the same needs and similar digital key situations over several weeks, and those whose needs and experiences varied more strongly over the weeks. Although "playing video games" was most frequently mentioned, individual analyses revealed that this was mainly due to multiple mentions over several weeks by single children. Adjusted for multiple mentions, the third category, "online communication or content sharing", was mentioned by most schoolchildren. It was shown in several studies that communication is a central motive for the use of digital media among children and adolescents [93–95]. In our study, this finding is further nuanced by the revelation that not only is communication a central motive for schoolchildren when using media, but moments involving computer-mediated communication seem to be so special or memorable that they are recalled from memory as key situations by many schoolchildren.

Key situations offer an important educational opportunity to use them for learning purposes [33]. Media competencies can thus be introduced close to the lives of schoolchildren, as they can present their own topics [31] and teachers can encourage reflection in a targeted manner. We found a wide range of different motives and topics that schoolchildren consider particularly important. This highlights the necessity for teachers to address the individual needs of schoolchildren, not only as a complement to established programs for learning media competencies but also to tap into the significant potential that arises from understanding the unique key situations experienced by each child. In particular, it should be noted that considerable variance in behavior was shown at the individual level, reflecting the individual personalities of the children [96], with certain topics being more personally relevant. In fact, previous findings have already shown that, in line with UAG, personality characteristics are correlated with preferences for certain news topics [97]. For example, individuals with high extraversion tend to use networking sites, instant messaging, and video chats more frequently [98]. More extraverted individuals share more information on social media, whereas individuals with high agreeableness and conscientiousness share less [99]. Particularly, individuals with high neuroticism are prone to problematic social media consumption, such as overuse and phubbing [100]. Similarly, the age of children can also impact media usage. For instance, a large study with over 9,000 adolescents aged 13 to 17 [101] found that older adolescents were more likely to use all social media platforms and reported greater exposure. However, younger adolescents used TikTok more than older adolescents. These findings align with results from the United States, where a study showed that older children (13–18 years) had more screen time compared to younger children (8–12 years) [102]. Furthermore, older age groups are more likely to exhibit problematic media consumption patterns than younger ones [103]. Also, socioeconomic factors like urban location, service occupations, and ethnicities influence media choices and usage [104]. Among children, a low socioeconomic status, lower socioemotional competencies, and lower intelligence are associated with higher media usage [105]. In Spain, schoolchildren from private schools spent less time on screen media compared to those attending state schools [106]. In our study, we did not find any significant differences between younger children (10–13 years) and older children (14–17 years) regarding the nature of key situations, associated emotions, communication, or media usage. Although personality traits, socioeconomic status, and cultural background play a crucial role in shaping media usage, they were not within the scope of our current study. Subsequent research should investigate the relationship between key situations and these factors to further elucidate their impact on media consumption behaviors.

Our category system provides an opportunity for further exploration with schoolchildren. The categories found can serve as a foundation for designing a quantitative survey to systematically investigate the prevalence and interrelations of these motives from key situations identified qualitatively. This sequential approach allows for a comprehensive understanding of the nuanced dynamics at play in children's media usage, facilitating evidence-based interventions and educational strategies tailored to their specific needs.

## 4.2 The setting in which key situations take place

Gaining an understanding of the differences and similarities in children's digital media usage within and outside of school is important. Educators have long favored for bridging the gap between formal learning environments and children's extracurricular activities to enrich their cognitive development [24, 25]. Our findings revealed that about 80% of all key situations occurred outside of the school. Furthermore, we observed that schoolchildren experienced a greater variety of key situations outside of school, while reporting more conventional experiences with Office documents and note-taking during classes. This disparity aligns with the findings of Lu et al. [107], who noted a more active and diverse use of social media by students outside of school compared to within it. Similarly, Clark et al. [108] discovered that teenagers engaged more extensively with social media outside of school than within it. However, Lu et al. [107] also highlighted similarities between these settings, indicating that students in both settings consume, share, and create content on social media platforms. Moreover, we found strong similarities in the results to another study we conducted with children of the same age [83]. In that study, schoolchildren used digital media more frequently outside of school for communication and entertainment than in class. Only regarding the motive of 'searching for information', we found no difference between the two settings. This is also consistent with our present study, where half of the reported key situations related to information search occurred in class, while the other half occurred outside of school. One plausible explanation for the variations in digital media usage between in-class and out-of-school settings in many areas is the varying degree of autonomy in fulfilling media-related needs. While teachers typically dictate the use of digital media within school premises, schoolchildren have more freedom to satisfy their media-related needs outside of school. Additionally, the integration of media in educational settings may be hindered by inadequate teacher training [26].

Although more than a third of all schoolchildren experienced at least one key situation during class on an individual level, our study confirms previous findings that media usage significantly varies between these two settings, with outside-of-school settings being preferred. It is noteworthy that the data for this study were collected at a time when, in the aftermath of the COVID-19 pandemic, schools and educational institutions were actively striving to create new opportunities for both schoolchildren and educators to engage with innovative teaching and learning methods [109]. This period saw an increased incorporation of media into educational practices [110]. Consequently, it is plausible that more key situations occurred within schools during the study period than before or after the pandemic. However, the utilization of digital media in classrooms remains highly dependent on the individual teacher's willingness to incorporate it into their instruction, and it is not yet standardized or internationally implemented in Germany (cf. [111]). This highlights significant potential for educators. Teachers could play a pivotal role in creating important key situations within the school setting during class time. By designing projects or assignments that align with schoolchildren's interests and needs, they can enhance their engagement and motivation. Providing more flexibility with media usage and addressing broader topics such as video games could also be beneficial (cf. [82, 112]). Allowing schoolchildren more freedom to explore various media platforms and

integrating topics that resonate with their experiences can make learning more relevant and meaningful. Furthermore, teachers could specifically address key situations experienced by schoolchildren outside of school and integrate them into the classroom, bridging the gap between informal and formal learning settings. By enabling schoolchildren to actively bring and reflect on their out-of-school experiences, educators can make the learning process more practical and relevant to schoolchildren's lives [33]. Through these measures, teachers could not only increase the relevance of the curriculum to schoolchildren's lives but also foster their digital literacy and support holistic education.

## 4.3 Emotional experiences during key situations

Our study results provide insights into the emotional experiences of schoolchildren in relation to their key situations with digital media. We found two important results: First, the majority of schoolchildren reported key situations with positive valence and low arousal, and second, there was no difference in intensity between key situations in the classroom and outside of school regarding emotional experiences. Additionally, we found no correlation between valence and arousal.

The first finding is consistent with the Mood Management Theory [57] which posits that individuals often engage with digital media to regulate their emotions, seeking content that enhances or sustains positive affective states. However, contrary to this general trend, a variability in emotional experiences emerged at the individual level. While some children only experienced positive affect with digital media over the entire survey period, for others this varied over the weeks, including negative affective states. Particularly notable were instances where challenges in utilizing digital media platforms elicited adverse emotional responses, specifically in the context of communication apps like WhatsApp. This variability underscores the nuanced interplay between individual differences and the emotional regulation strategies employed in media consumption, in accordance with the principles outlined in the Mood Management Theory. Other studies have also shown negative feelings associated with the use of WhatsApp and similar communication apps. Children are increasingly exposed to bullying and harassment on WhatsApp, which can have serious negative consequences [113]. Additionally, children often face privacy issues and encounter unregulated messages on WhatsApp, leading to exposure to inappropriate content and potential security risks [114].

Furthermore, our findings reveal that certain key situations prompted heightened levels of arousal among the participants, particularly during moments of pronounced valence. This observation corroborates the theory's assertion that individuals may seek out media content not only to modulate their mood but also to experience heightened emotional states, thereby enhancing their overall affective experiences. An explanation for the multitude of positive emotions associated with key situations may lie in the situations themselves, which had direct positive effects on the emotions of the schoolchildren. The experiences that arise during the use of digital media go beyond mere mood regulation and could have provided profound positive experiences. By acquiring new skills or realizing creative projects, schoolchildren may have experienced a sense of competence and self-efficacy, which in turn can lead to an increased sense of satisfaction and well-being [115]. Worth mentioning in the context of experienced valence and arousal, there was no significant correlation between these two aspects, which is consistent with theoretical models emphasizing the independence of valence and arousal [64, 65].

The second main finding suggests that the emotional significance of key situations remains similar regardless of the setting. Emotions experienced may be more influenced by individual experiences or the subjective perception of the situation itself rather than its location or

contextualization. This observation may indicate that schoolchildren exhibit similar emotional responses to various types of key situations, whether they occur in a school or non-school setting. This result was also found in our other study [83], where we discovered that schoolchildren are equally satisfied with their use of digital media in both settings. It may be interesting to further analyze the nature of these key situations in future studies to determine specific aspects that lead to their positive evaluation by schoolchildren, regardless of the setting.

When examining the individual key situations (Fig 1), it is noteworthy that key situations involving gaming consoles were almost exclusively associated with positive valence and high arousal. This is particularly striking given that, in general, arousal levels were relatively low. However, this finding aligns with current research, which suggests that video games tend to increase physiological arousal [116, 117]. Notably, active gameplay results in a greater increase in arousal compared to passive watching [117]. For computers and smartphones, there was more variability in both arousal and valence, likely due to the greater heterogeneity of key situations in this category. Particularly low arousal was associated with tablet usage.

Our findings have practical implications for teachers and parents, as they should be aware of how emotions can influence their schoolchildren' media use. By understanding the emotional reactions, they can respond better and take supportive measures. Teachers can leverage schoolchildren' emotions to make their teaching more engaging and promote positive emotions. This may include integrating digital media and activities that increase their emotional engagement. If schoolchildren experience negative emotions during media use, it is important for teachers and parents to provide them with supportive resources and strategies to cope with these emotions [42]. Particularly with communication apps, there is a need to be mindful of content that might provoke negative emotions. In this area, parents and teachers play a key role in fostering a balanced and informed approach, as overly restrictive regulation by parents can sometimes lead to increased problematic use [118].

## 4.4 Post-communication about key situations and social support

The present study examined the communication patterns of schoolchildren regarding digital key situations and the associated social support. Findings indicate that the majority of schoolchildren actively sought communication about their experiences. This underscores the importance of exchanging key situations for children [31, 33]. Particularly, direct face-to-face communication was preferred, while online communication methods were notably less utilized.

The results suggest that family members are the primary conversation partners for children, followed by school friends and less frequently, school staff. This supports the assumption that close relationships, especially within the family and among peers, are crucial for children's development [71, 74, 119]. The highest intensity of conversations was observed with family members, indicating that children experience a high level of trust and support within this group. Interestingly, there was a lower use of online communication channels for exchanging key situations. While this contrasts with the growing importance of digital communication in other areas of life [75], it aligns with the findings of Zimmermann et al. [74]. This discrepancy suggests that personal interactions are still preferred for processing and exchanging personal experiences, particularly when it comes to sensitive topics and key situations.

Regarding perceived social support, a diversity of experiences was evident among the schoolchildren. While some children experienced strong support, others felt less supported. These differences could stem from individual variations in their relationships with conversation partners as well as the nature of the key situations, highlighting the importance of the social environment in supporting children across various domains of life [69].

The findings provide significant implications for both practice and research in the field of education. Given that school staff were utilized as conversation partners, but notably less frequently than parents and friends, education institutions could offer training sessions and workshops to support teachers in developing effective communication strategies. This initiative could enhance teachers' abilities to address the needs of children and foster a supportive environment. While the preference for face-to-face communication is evident, schools should still consider integrating digital communication methods into their daily routines. This could involve establishing secure online platforms for information exchange and support, catering to the needs of schoolchildren who may not always prefer personal communication. Additionally, it is crucial to conduct further research to investigate the effectiveness of various communication strategies in schools. This research could include evaluating interventions aimed at promoting communication and social support to determine which approaches are most effective.

## 4.5 Reflection on key situations

Reflection on key situations is essential for schoolchildren's learning and growth, as highlighted by various theoretical perspectives such as Kolb's experiential learning model [79]. Our results show that a significant portion of schoolchildren reported they would not change anything in hindsight regarding their key situations. However, a notable percentage indicated that they would do something differently next time, demonstrating their capacity for self-reflection and improvement. These reflections cover a range of themes, including aspects of media consumption, time management, interpersonal interactions, and technical skills, highlighting the diverse nature of their learning experiences. Furthermore, many schoolchildren reported they learned something from their key situations, emphasizing the educational value of these digital experiences. Their learning extends beyond acquiring substantive knowledge to include self-awareness, digital literacy, and personal growth. This underscores the importance of providing opportunities for schoolchildren to reflect on their digital interactions and derive meaningful insights from them, see [80, 81].

Furthermore, these findings hold practical implications for schools. Educators can leverage the understanding derived from schoolchildren's reflections on key situations to enrich everyday educational practices. Incorporating opportunities for dialogues with schoolchildren about their key situations can enhance teacher-student relationships and create a supportive learning environment [33]. By actively engaging in conversations about digital experiences, educators can gain valuable insights into schoolchildren's perspectives, concerns, and learning needs. As Rüth and Kaspar [82] demonstrated, incorporating video games into school lessons is an effective way to stimulate reflection. This approach successfully integrates the lifeworld of schoolchildren with formal teaching.

## 4.6 Hardware and software

The schoolchildren used smartphones most frequently in their key situations. This aligns with findings that 94% of children between the ages of 12 and 19 in Germany own a smartphone [93]. Efforts are already being made to bridge informal and formal settings through concepts like BYOD, where schoolchildren can bring their own smartphones into the classroom. Since they are more familiar with their own devices, the learning experience can be enhanced, and motivation to learn increases [120]. Although television still plays a central role for children [93], this medium was significantly less prominent in our key situation findings, possibly due to its limited interactivity [99], which may result in less deeply processed experiences [121]. Only two key situations involved the use of more than one digital medium, suggesting media multitasking, which is often associated with negative effects [122, 123].

According to UGA [34], preferred media choices tend to be based on the extent to which they satisfy users' specific needs and gratifications. For example, smartphones are favored for their portability and instant service access, while tablets and computers are chosen for in-depth information seeking and content creation [124]. Our data reflect similar trends; for example, key situations related to "Online communication or content sharing" were almost exclusively experienced on smartphones, with only three cases involving other media. Conversely, in the categories "Creating, editing, or presenting Office documents" and "Acquiring digital media competencies," smartphones were used in just one instance. For other categories, no clear preference emerged across key situations, likely because tasks today can be managed on nearly any device, leaving individual preferences to determine gratification.

On an individual level, a pattern emerges where some schoolchildren seem to favor a particular digital medium for experiencing key situations. Many schoolchildren who reported four key situations consistently experienced them using the same medium across all four weeks (e.g., Child 2, 3, 5). Others showed a tendency towards one medium, even if they did not exclusively report using it. For example, Child 4 experienced three key situations with a smartphone and one with a computer. Determining the extent to which this reflects a fixed pattern or is influenced by situational or personality factors warrants further research.

Regarding software, multiple responses were possible, making it unclear what proportion of specific programs or websites were used. For example, in one key situation, where it was reported that a presentation for class was created (Category No. 4), it was stated that "Power-Point," "Google," and "Wikipedia" were used. Google emerged as the most frequently mentioned search tool, although a significantly smaller number of reported key situations could be assigned to the category "Searching for information on the internet". This indicates that schoolchildren may search for information casually without consciously noting it. WhatsApp, the second most commonly mentioned tool, aligns with findings that "Online communication or content sharing" was one of the most frequently reported key situations. Importantly, given the young age of the participants, WhatsApp class groups warrant caution, as these may expose children to issues such as cyberbullying [125], sexting, or viral videos with negative impacts [126]. Thus, the role of parents and teachers in guiding WhatsApp usage is essential. Restrictive regulation strategies may at times lead to problematic use, underscoring the need for balanced and informed parental guidance [118].

In connection with hardware and software availability, the digital divide also warrants consideration. The digital divide significantly affects schoolchildren, particularly those from lower socioeconomic backgrounds. Limited access to digital devices and educational software in high-poverty schools exacerbates this divide, creating barriers to digital literacy and impacting learning opportunities as schoolchildren progress through school [127]. Such disparities can hinder children's ability to gain the same digital media competencies and experience the same breadth of key situations as their peers in better-resourced schools. This access gap highlights the importance of equitable access initiatives to ensure all schoolchildren can develop the necessary skills to navigate digital environments effectively and responsibly.

### 4.7 Limitations

Finally, we want to highlight some limitations of the present study and associated prospects for future research.

First, the sample was obtained through convenience sampling, potentially comprising only particularly motivated schoolchildren. Moreover, the study was conducted in a single federal state in Germany, although it included multiple schools and different types of schools, enhancing diversity. As we focused strongly on the explorative investigation of the qualitative facets of

key situations in this study, we did not conduct a priori power analysis required for more in-depth quantitative analyses. However, despite the challenges posed by the COVID-19 conditions, we aimed to achieve the largest possible sample size with this hard-to-reach group. The small sample size is consistent with other qualitative studies in the field of media research (e.g., [128–130]) and electric diaries [131], where a smaller, purposefully selected sample is often used to gain deeper insights into specific social phenomena [132]. Nevertheless, future research could aim to include larger and more diverse populations to potentially uncover differences between different nationalities or locations. For instance, considering the usage gap hypothesis [133], which posits that individuals from different socioeconomic backgrounds have varying access to media offerings and hardware, the key situations experienced by children may differ significantly.

Second, there might be a bias in responses regarding key situations, with a tendency to report predominantly positive experiences due to social desirability bias. It is unclear whether schoolchildren truly experienced mostly positive key situations or simply reported them as such. However, the potential for bias was partially reduced by conducting an online survey, which might have promoted greater openness and honesty, thus diminishing the influence of social desirability response bias [134–136]. Nevertheless, for future research, it might be conceivable to inquire about both a positive and a negative key situation within a week.

Third, due to the study design, it is possible that the schoolchildren reported more key situations outside of school, as access to the study was available from Friday to Sunday evening. This could suggest a recency effect. On the other hand, we specifically aimed to ask about the one moment that stood out most in their memory, which is why we could only open the survey at the end of the week. This ensured that the children would not participate at the beginning of the week and then realize by the end of the week that a more significant situation had occurred. Our definition of a key situation includes the requirement that it is particularly important and, as such, is well remembered. If a situation is forgotten within a few days, it does not meet our definition.

Last, the quantitative assessment of emotions may have limitations. It is unclear exactly how emotions were experienced during the key situation, including whether there were mixed feelings or if emotions changed over time. Future research could explore emotions through qualitative self-reports rather than using a scale, allowing for a more nuanced understanding of the type of schoolchildren's emotional experiences.

## 5. Conclusions

In conclusion, our study provides valuable insights into digital key situations, settings, emotional experiences, communication patterns, and reflections of schoolchildren in their digital media usage. By examining these aspects comprehensively, we have identified important implications for educators, parents, and researchers. Table 4 presents an overview of the main findings along with their theoretical and practical implications.

A key challenge in teaching digital literacy is that it is not yet an independent subject in many countries, including Germany [137]. Instead, it is treated as a cross-cutting theme, and its integration into individual subjects depends largely on the approach taken by the teacher. This makes it difficult to systematically and consistently embed media education into the curriculum, which in turn limits its potential to foster student motivation and individual development. To address this, educators can leverage key situations as opportunities for learning and skill development, bridging the gap between formal education and schoolchildren's digital experiences. By integrating these experiences into the classroom and fostering reflection, teachers can enhance the relevance and effectiveness of educational interventions. Parents play

**Table 4. Summary of the key findings and implications of this study.**

| | Key findings | Theoretical implications | Practical implications |
|---|---|---|---|
| Key situations in media use (RQ1) | We identified 14 categories of key situations. | Our category system was established following the methodology outlined by Mayring [89] and exhibited a high kappa value. Based on this, the category system can serve as a foundation for further (representative) research in the field. | Teachers can address the individual needs of schoolchildren by incorporating their unique key situations into learning activities. In particular, incorporating video games into classroom instruction has great potential to boost schoolchildren's motivation. |
| | Most key situations were categorized as "Playing video games". Most schoolchildren mentioned topics from the category "Online communication or content sharing". | | |
| | At the individual level, key situations varied significantly among schoolchildren. | The findings suggest that individual differences in media usage could be linked to various motivational or personality factors. This implies that different personalities have different interests and focal points, which can influence how they engage with digital media. Furthermore, personality, socioeconomic status, and cultural factors can also influence media usage. | The variation in key situations suggests that a one-size-fits-all approach may not be effective. Teachers can design a more inclusive curriculum that considers the diverse interests and experiences of their schoolchildren, thereby promoting a more equitable learning environment. |
| Setting of key situations (RQ2) | There was much more diversity in key situations outside of school. | The findings highlight the need for a deeper understanding of the differences and similarities in children's digital media usage within and outside of school. This can inform theories on how varying degrees of autonomy and teacher guidance impact media-related behaviors and learning experiences. | Teachers can bridge the gap between informal and formal learning environments by integrating key situations experienced by schoolchildren outside of school into the classroom. This approach can make the learning process more practical and relevant, fostering schoolchildren's digital competence and supporting holistic education. |
| | Many categories were not reported in school, such as key situations related to video games, videos, or communication. | The lack of reported key situations related to certain media categories in school suggests that formal educational settings might limit the diversity of media experiences. | Digital media are fast-paced and continually introduce new trends. Comprehensive teacher training can ensure that pre-service teachers are equipped to handle and keep up with these changes. In-service teachers can be kept up-to-date through regular professional development sessions. |
| Emotional experiences during key situations (RQ3) | Schoolchildren mainly reported key situations where they felt happy and had low arousal. However, there were also key situations where schoolchildren felt sad or experienced high arousal. | Either the positive key situations were better remembered by the schoolchildren, or there was a bias causing them to prefer reporting the happy moments in the study. A change in the study design could bring clarity in the future. | When schoolchildren experience negative emotions during media use, teachers and parents should provide supportive resources and coping strategies to help them manage these feelings effectively. |
| Post-communication about key situations and social support (RQ4) | In over 70% of key situations, schoolchildren talked to others about the experience afterwards. | The high percentage of schoolchildren discussing key situations with others reinforces the idea that social support and communication are vital components of child development. | Schoolchildren frequently talk about their key situations with others, providing teachers with an opportunity to address these experiences. Here, they can actively engage with the schoolchildren. |
| | They most frequently and intensely talked with family members, followed by school friends. They spoke less often with school staff and even less with non-school friends. | The findings support existing theories on the importance of close relationships, particularly with family members and peers. | Schools and universities could offer training sessions and workshops to help (pre-service and in-service) teachers develop effective communication strategies, enhancing their ability to address children's needs and foster a supportive environment. |
| | Schoolchildren rarely used online communication and almost always discussed key situations face-to-face. | The study highlights the preference for face-to-face communication over online methods among schoolchildren, suggesting that personal interactions are crucial for processing key situations. | Teachers can educate schoolchildren regarding online communication and discuss differences and similarities together, enabling them to competently engage in online interactions. |
| | In more than half of the key situations, schoolchildren sought social support and received varying levels of it. | The diversity of perceived social support among schoolchildren suggests that individual differences in relationships and the nature of key situations significantly influence the level of support experienced. This variability underscores the need for theoretical frameworks to account for the nuanced impacts of social environments on children's development. | It is unclear from whom the schoolchildren received social support from and where there is still potential for improvement. Nevertheless, teachers can draw from this that schoolchildren can be more engaged in their valuable experiences. |

*(Continued)*

**Table 4.** (Continued)

| | Key findings | Theoretical implications | Practical implications |
|---|---|---|---|
| Reflection on key situations (RQ5) | In a quarter of the key situations, schoolchildren expressed a desire to do something differently in hindsight. What they would like to do differently strongly depended on the key situation. | When schoolchildren express a desire to do something differently, this suggests that they are capable of evaluating their own actions and recognizing alternative courses of action. | Educators can leverage schoolchildren's reflections on key situations to enhance everyday educational practices. Incorporating opportunities for dialogue about digital experiences can enrich teacher-schoolchild relationships and create a supportive learning environment. |
| | In slightly more than half of the key situations, schoolchildren indicated that they had learned something. The statements can be divided into three contexts: they learned something about themselves, about the digital medium, or gained substantive knowledge. | This finding supports theoretical perspectives such as Kolb's experiential learning model (1984), emphasizing the importance of reflection in the learning process. | At this juncture, it's crucial for teachers to intervene and bolster processes of reflection. This way, digital competence can be learned firsthand. |

a crucial role in supporting their children's digital media usage by providing a supportive environment for communication and social support. By engaging in open dialogues and understanding their children's emotional experiences, parents can promote healthy media habits and emotional well-being. Researchers can build upon our findings by conducting further studies to address the limitations highlighted and explore new avenues for understanding schoolchildren's digital media behavior. By adopting interdisciplinary approaches and longitudinal designs, researchers can deepen our understanding of the complex interplay between digital media usage, emotional experiences, and social dynamics among schoolchildren. Overall, our study contributes to the growing body of literature on children's digital media usage and underscores the importance of considering the diverse needs, experiences, and contexts of schoolchildren in the digital age. Through collaborative efforts between educators, parents, and researchers, we can create supportive environments that empower schoolchildren to navigate digital media responsibly and maximize the educational opportunities afforded by these technologies.

## Supporting information

**S1 Table. Expression of the Big Five traits of all school children.**
(DOCX)

## Author Contributions

**Conceptualization:** Jennifer Virginie Meier, Kai Kaspar.

**Data curation:** Jennifer Virginie Meier, Kai Kaspar.

**Formal analysis:** Jennifer Virginie Meier.

**Funding acquisition:** Kai Kaspar.

**Investigation:** Jennifer Virginie Meier.

**Methodology:** Jennifer Virginie Meier, Kai Kaspar.

**Project administration:** Jennifer Virginie Meier, Kai Kaspar.

**Resources:** Kai Kaspar.

**Supervision:** Kai Kaspar.

**Validation:** Jennifer Virginie Meier, Kai Kaspar.

**Visualization:** Jennifer Virginie Meier.

**Writing – original draft:** Jennifer Virginie Meier.

**Writing – review & editing:** Kai Kaspar.

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
