## [Decision Letter · Decision Letter 0]

31 Oct 2024

PONE-D-24-44716Revealing schoolchildren’s key situations in the use of digital media inside and outside school – a media diary studyPLOS ONE

Dear Dr. Meier,

Thank you for submitting your manuscript to PLOS ONE. After careful consideration, we feel that it has merit but does not fully meet PLOS ONE’s publication criteria as it currently stands. Therefore, we invite you to submit a revised version of the manuscript that addresses the points raised during the review process.

Please submit your revised manuscript within Dec 15 2024 11:59PM. If you will need more time than this to complete your revisions, please reply to this message or contact the journal office at plosone@plos.org. Please include the following items when submitting your revised manuscript:A rebuttal letter that responds to each point raised by the academic editor and reviewer(s). You should upload this letter as a separate file labeled 'Response to Reviewers'.A marked-up copy of your manuscript that highlights changes made to the original version. You should upload this as a separate file labeled 'Revised Manuscript with Track Changes'.An unmarked version of your revised paper without tracked changes. You should upload this as a separate file labeled 'Manuscript'.If applicable, we recommend that you deposit your laboratory protocols in protocols.io to enhance the reproducibility of your results. Protocols.io assigns your protocol its own identifier (DOI) so that it can be cited independently in the future. For instructions see: https://journals.plos.org/plosone/s/submission-guidelines#loc-laboratory-protocols. Additionally, PLOS ONE offers an option for publishing peer-reviewed Lab Protocol articles, which describe protocols hosted on protocols.io. Read more information on sharing protocols at https://plos.org/protocols?utm_medium=editorial-email&utm_source=authorletters&utm_campaign=protocols.

We look forward to receiving your revised manuscript.

Kind regards,

Mohamed Ahmed Said, Ph.D.

Academic Editor

PLOS ONE

Journal Requirements:

Reviewers' comments:

Reviewer's Responses to Questions

**Comments to the Author**

1. Is the manuscript technically sound, and do the data support the conclusions?

Reviewer #1: Partly

Reviewer #2: Yes

Reviewer #3: Yes

Reviewer #4: Yes

Reviewer #5: Yes

2. Has the statistical analysis been performed appropriately and rigorously? 

Reviewer #1: I Don't Know

Reviewer #2: Yes

Reviewer #3: Yes

Reviewer #4: Yes

Reviewer #5: Yes

3. Have the authors made all data underlying the findings in their manuscript fully available?

Reviewer #1: Yes

Reviewer #2: Yes

Reviewer #3: Yes

Reviewer #4: Yes

Reviewer #5: Yes

4. Is the manuscript presented in an intelligible fashion and written in standard English?

Reviewer #1: Yes

Reviewer #2: Yes

Reviewer #3: Yes

Reviewer #4: Yes

Reviewer #5: Yes

5. Review Comments to the Author

Reviewer #1: This is a very interesting article. The author studied the media usage of 49 students aged 10-17 n in both educational and non-educational, which has become a common phenomenon worthy of our collective research. However, there are some issues with the article that need further clarification:

1. What are the basic characteristics of the 8 schools where the research samples were collected? Are they of the same type, public or private schools? What is the scale of these schools, and how diverse is the student population in terms of individual differences?

2. Can the sample size of 49 support the research conclusions? How were these 49 samples selected? Is there any reference material to back up the sample size?

3. Considering the psychological characteristics of students aged 10-17, could these factors lead to different conclusions regarding media usage?

4. The article has sufficient data analysis in the Results section, but the academic depth of the analysis for the conclusions is lacking.

5. The author's conclusion that "115 key situations occurred outside of school, while only 30 occurred in class" likely reflects the genuine experiences of most students, as media use in schools is mainly for learning activities, while it gets more complex outside of education. My question is: since this conclusion seems very commonsensical, where does the significance and innovation of the research lie?

Reviewer #2: This study is excellent and addresses a critical topic. I thank the authors for their effort and for providing the opportunity to read such a valuable piece of work. I do not see the need for significant revisions; only a few suggestions could further enrich the study.

1. The authors need to clarify specific details related to the study sample. For instance, it would be helpful to know the number of hours in the school day and whether there are any restrictions or regulations on students bringing digital devices (such as mobile phones or tablets) to school. There is a considerable difference between digital activities conducted inside and outside the classroom, and understanding the reasons for this variation would be beneficial.

2. Regarding young students (10 years old) interacting in WhatsApp groups, how are children of this age engaging with such applications? Do parents monitor these groups?

3. Regarding children participating in activities from Friday through Monday, I believe this period might favor recalling activities outside school. Tuesday through Thursday is a relatively long stretch for students to remember in-school activities. The authors should clarify why children could not record their weekly activities all the week’s days.

4. Lastly, focusing on the differences attributed to age in the results would enhance the discussion, as significant age-related variations exist within the study sample.

Reviewer #3: The manuscript meets the requirements according to the publication criteria. It has a sophisticated scientific language, a robust theoretical argumentation, in addition to important analyses according to quantitative and qualitative methodology.

It is mentioned in the text that it is a qualitative manuscript, however, it presents many quantitative analyses, so my recommendation would be to present it as a mixed concurrent research.

It is true that the sample size is small, but it is mentioned in the limitations as an element to be taken into account.

Reviewer #4: I would like to thank you for the opportunity to review the paper titled Revealing Schoolchildren’s Key Situations in the Use of Digital Media Inside and Outside School – A Media Diary Study. This research project addresses an important topic and is well-written.

Overall, the study provides valuable insights into schoolchildren's media usage, grounded in established theories such as the Uses and Gratifications approach. The article presents significant implications for both school practice and academic research. Notably, it is a unique study in its effort to examine children’s digital media usage across both formal and informal settings. The researchers also offer valuable theoretical and practical contributions. Based on these aspects, the manuscript makes a meaningful addition to the literature on digital media use among schoolchildren.

However, I would like to suggest a few considerations should this article be deemed suitable for publication in your journal:

1. You should consider the journal's policy regarding the small sample size of the study, as only 49 schoolchildren were included in the analysis. Additionally, 22 schoolchildren were classified as dropouts (line 152), resulting in a response rate of approximately 70%. I recommend that the researchers address the high dropout rate, perhaps by analyzing demographic differences between the participating schoolchildren and dropouts (e.g., gender, school affiliation, grade).

2. In the first part of the introduction, I recommend adding a detailed definition of the components of 'digital media' for the various everyday purposes mentioned in line 36 (outside the classroom). This may include specifying the use of 'news media' through traditional online and social media platforms, other social interactions through social media, and additional digital media outlets such as office tools, search engines, video games, streaming entertainment, music, etc. If possible, relevant findings on different types of digital media could also enhance the analysis, especially by differentiating news media consumption from entertainment and documentation, which may offer a broader scope.

3. Did the study also include demographic data on the families-parents of the participating schoolchildren? Information such as parents' education level or family socioeconomic status could provide additional insights in interpreting the results.

4. Regarding the finding that key situations varied significantly among schoolchildren at the individual level, the researchers suggest that differences in media usage could be linked to various motivational or personality factors (Table 4, after line 596). The researchers might also consider other factors that could influence differences in media usage, such as socioeconomic and cultural factors.

5. I would suggest providing a more thorough definition of 'key situations' in the introduction section (line 43). The current definition is quite brief, and key situations are not limited to schoolchildren, and can also apply to adults. Additionally, I noticed a typo error in line 42: "s is".

6. A grammatical remark: I noticed in line 28 "digital media" is presented in singular ("has"), however in other sentences it is presented in plural (for example in lines 34, 48, 342) – there should be uniformity, apparently in plural since the definition reflects several digital media outlets.

Sincerely.

Reviewer #5: Dear Authors, I hope you find in a good health. First of all I would like to congratulate for the research.

there are some comments to improve the text:

Strengths:

Relevant and current topic, considering the increasing use of digital media by children and adolescents; Solid qualitative methodology, using media diaries to capture key situations; Detailed discussion of the results and emotional implications of media use.

Areas for improvement:

The relatively small and limited sample size may limit the generalizability of the results; The theoretical section could be more robust, including more references and theoretical models beyond the UGA; Longitudinal analysis of the data and more statistical analyses could strengthen the conclusions; Practical implications for educators could be more detailed.

1. Title and Abstract: The title is clear and reflects the objective of the study; The abstract is well structured and covers the main points of the study. However, it is possible to simplify some sentences to make the reading more fluid, keeping the focus on the main findings.

2. Introduction: The introduction provides a good background on children’s use of digital media in educational and non-educational contexts. However, I suggest including a clearer justification for the relevance of studying the difference in media use across these contexts.

3. Methodology: The choice of the media diary as a qualitative method is appropriate, but it would be important to include more details on how the categories for qualitative analysis were developed; The relatively small number of participants (n=49) could be a limitation.

4. Results: The presentation of the results is clear and well organized, especially regarding the analysis of emotional variation (valence and arousal). It could benefit from greater contextualization with previous studies on the emotional impact of media use by children and adolescents. Suggestion: for data visualization: Including graphs or infographics to illustrate key trends, such as the most used type of media.

5. Discussion is ok

6. Conclusion: The conclusion summarizes the findings effectively, but could benefit from more direct suggestions for educational practice. For example, how schools can integrate these findings to improve classroom use of digital media in a balanced way with out-of-school use.

6. PLOS authors have the option to publish the peer review history of their article (what does this mean?). If published, this will include your full peer review and any attached files.

Reviewer #1: No

Reviewer #2: No

Reviewer #3: **Yes: **Dr. Jose Manuel Meza Cano

Reviewer #4: No

Reviewer #5: **Yes: **Vinicius Barroso Hirota, PhD.

---

## [Author Response · Author response to Decision Letter 0]

23 Nov 2024

Thank you for your time and effort in reviewing our manuscript. We sincerely appreciate the valuable feedback and constructive suggestions provided by all five reviewers. These comments have significantly contributed to improving the quality and clarity of our work.

All responses to the reviewers' suggestions have been addressed and are included in the attached "Response to Reviewers" document.

---

## [Decision Letter · Decision Letter 1]

13 Dec 2024

Revealing schoolchildren’s key situations in the use of digital media inside and outside school: a media diary study

PONE-D-24-44716R1

Dear Dr.Jennifer Virginie Meier,

We’re pleased to inform you that your manuscript has been judged scientifically suitable for publication and will be formally accepted for publication once it meets all outstanding technical requirements.

Kind regards,

Mohamed Ahmed Said, Ph.D.

Academic Editor

PLOS ONE

Additional Editor Comments (optional):

Reviewers' comments:

Reviewer's Responses to Questions

**Comments to the Author**

1. If the authors have adequately addressed your comments raised in a previous round of review and you feel that this manuscript is now acceptable for publication, you may indicate that here to bypass the “Comments to the Author” section, enter your conflict of interest statement in the “Confidential to Editor” section, and submit your "Accept" recommendation.

Reviewer #1: All comments have been addressed

Reviewer #2: All comments have been addressed

Reviewer #4: All comments have been addressed

Reviewer #5: All comments have been addressed

2. Is the manuscript technically sound, and do the data support the conclusions?

Reviewer #1: Yes

Reviewer #2: Yes

Reviewer #4: Yes

Reviewer #5: Yes

3. Has the statistical analysis been performed appropriately and rigorously? 

Reviewer #1: Yes

Reviewer #2: Yes

Reviewer #4: Yes

Reviewer #5: Yes

4. Have the authors made all data underlying the findings in their manuscript fully available?

Reviewer #1: Yes

Reviewer #2: Yes

Reviewer #4: Yes

Reviewer #5: Yes

5. Is the manuscript presented in an intelligible fashion and written in standard English?

Reviewer #1: Yes

Reviewer #2: Yes

Reviewer #4: Yes

Reviewer #5: Yes

6. Review Comments to the Author

Reviewer #1: It is suggested to further improve the paper in accordance with plos one specifications, accurately express the research conclusions, so that readers can more introduce and obtain the research results.

Reviewer #2: Congratulations on the great job you did with your work. It was truly impressive, and I admire the effort and dedication you put into it. I hope this is just the beginning of more success for you, and I look forward to seeing your future projects flourish. Wishing you all the best in your upcoming endeavors!

Reviewer #4: I would like to re-express my gratitude for the opportunity to review the paper titled "Revealing Schoolchildren’s Key Situations in the Use of Digital Media Inside and Outside School – A Media Diary Study". This research project addresses a significant and timely topic and is well-written.

After carefully reviewing the revised manuscript, I found that the researchers have effectively addressed the concerns raised by the reviewers and made appropriate additions to clarify the research topics. Therefore, I'm pleased to recommend the manuscript for publication.

Respectfully yours

Reviewer #5: Dear Author(s), I hope you find in a good health, and please, consider the review to improve the best publication for you.

1. General Structure and Clarity

Title: Both documents have clear and similar titles. However, in the revised document, there is better detailing of the keywords and a more cohesive presentation of the research questions.

Abstract: The abstract in the revised document is more complete, adding clarity about the use of mixed methods and the most significant findings. It also emphasizes the practical implications more explicitly.

2. Introduction and Theoretical Foundation

Additional Citations: The revised document includes more references to support its claims, adding credibility to statements about the impact of digital media on children.

3. Methodology

Process Details:

The revised document includes more detailed descriptions of the data collection process and the use of mixed methods.

Improved explanations of how qualitative coding was conducted (e.g., iterative approaches and inter-coder reliability).

Data Analysis: The revised version provides a clearer justification for choosing descriptive analyses and includes references to validate the chosen approach.

4. Results

Clearer Presentation: In the revised document, the categories of "key situations" are better organized, with examples and tables explained in greater detail. There is an effort to make the results more accessible through more direct descriptions.

Additional Emotional Details: The discussion of children's emotional experiences during key situations is expanded, integrating concepts of "valence" and "arousal" in more depth.

5. Discussion

Practical Implications:

The discussion in the second manuscript provides more recommendations for educators, indicating how the findings can be applied in school settings.

The revised version better addresses limitations (e.g., sample size) and suggests future directions for research.

6. Limitations and Ethical Considerations

The revised version more explicitly acknowledges methodological limitations, such as the small sample size and challenges in generalization, which strengthens the study's credibility.

Overall Conclusion

The document shows significant improvements in terms of:

Clarity and theoretical depth.

Methodological explanation and justification of analytical choices.

Organization of results and practical relevance.

Congretulations.

7. PLOS authors have the option to publish the peer review history of their article (what does this mean?). If published, this will include your full peer review and any attached files.

Reviewer #1: No

Reviewer #2: **Yes: **Abdelmohsen Hamed Okela

Reviewer #4: No

Reviewer #5: **Yes: **Vinicius Barroso Hirota, PhD.

---

## [Editor Report · Acceptance letter]

18 Dec 2024

PONE-D-24-44716R1 

PLOS ONE

Dear Dr. Meier, 

I'm pleased to inform you that your manuscript has been deemed suitable for publication in PLOS ONE. Congratulations! Your manuscript is now being handed over to our production team.

Kind regards, 

on behalf of

Dr. Mohamed Ahmed Said 

Academic Editor

PLOS ONE